# Analysis of maternal and newborn training curricula and approaches to inform future trainings for routine care, basic and comprehensive emergency obstetric and newborn care in the low- and middle-income countries: Lessons from Ethiopia and Nepal

Gaurav Sharma[1]*, Yordanos B. Molla[2], Shyam Sundar Budhathoki[3], Million Shibeshi[4], Abraham Tariku[5], Adhish Dhungana[6], Bindu Bajracharya[7], Goitam G. Mebrahtu[8], Shilu Adhikari[9], Deepak Jha[10], Yunis Mussema[11], Abeba Bekele[12], Neena Khadka[2]

1 Society of Public Health Physicians, Kathmandu, Nepal, 2 USAID's Maternal and Child Survival Program/ Save the Children, Washington, DC, United States of America, 3 Nepalese Society of Community Medicine, Lalitpur, Nepal, 4 Independent Consultant, Addis Ababa, Ethiopia, 5 Federal Ministry of Health, Addis Ababa, Ethiopia, 6 USAID's Maternal and Child Survival Program/Save The Children, Kathmandu, Nepal, 7 Independent Consultant, Kathmandu, Nepal, 8 Black Lion Hospital, Addis Ababa, Ethiopia, 9 USAID Nepal, Kathmandu, Nepal, 10 Child Health Division, Ministry of Health and Population, Kathmandu, Nepal, 11 USAID Ethiopia, Addis Ababa, Ethiopia, 12 USAID's Maternal and Child Survival Program/Save The Children, Addis Ababa, Ethiopia

* drsharmag@gmail.com

## Abstract

Program managers routinely design and implement specialised maternal and newborn health trainings for health workers in low- and middle-income countries to provide better-coordinated care across the continuum of care. However, in these countries details on the availability of different training packages, skills covered in those training packages and the gaps in their implementation are patchy. This paper presents an assessment of maternal and newborn health training packages to describe differences in training contents and implementation approaches used for a range of training packages in Ethiopia and Nepal. We conducted a mixed-methods study. The quantitative assessment was conducted using a comprehensive assessment questionnaire based on validated WHO guidelines and developed jointly with global maternal and newborn health experts. The qualitative assessment was conducted through key informant interviews with national stakeholders involved in implementing these training packages and working with the Ministries of Health in both countries. Our quantitative analysis revealed several key gaps in the technical content of maternal and newborn health training packages in both countries. Our qualitative results from key informant interviews provided additional insights by highlighting several issues with trainings related to quality, skill retention, logistics, and management. Taken together, our findings suggest four key areas of improvement: first, training materials should be updated based on the content gaps identified and should be aligned with each other. Second, trainings should address actual health worker performance gaps using a variety of

**Data Availability Statement:** All relevant data are within the paper and its Supporting Information files.

**Funding:** This study was made possible by the generous support of the American people through the United States Agency for International Development (USAID), under the terms of Cooperative Agreement No. AID-OAA-A-14-00028. The contents are the responsibility of the authors and do not necessarily reflect the views of USAID or the United States Government.The funders had no role in study design, data collection and analysis, decision to publish, or preparation of the manuscript.

**Competing interests:** The authors have declared that no competing interests exist.

innovative approaches such as blended and self-directed learning. Third, post-training supervision and ongoing mentoring need to be strengthened. Lastly, functional training information systems are required to support planning efforts in both countries.

## Introduction

The health of pregnant women and neonates are closely aligned, and there is growing emphasis on the promotion of integrated delivery of services across the continuum of care for maternal, newborn and child health [1]. Health systems are weaker and resource limitations are more pronounced in LMIC settings which has considerable implications for the efficient delivery of quality health services [2, 3]. It is now well accepted that training alone may not be enough to bring lasting improvements to the quality of care without improving wider health systems issues such as availability of equipment and supplies, human resources, clinical governance mechanisms and environments [4]. However, trainings of health workers, either individually or in combination, are generally the first step undertaken by any program aiming to improve maternal and newborn health services [4–7].

Effective interventions for routine and emergency care for mothers and neonates are all well-established. Most maternal and newborn deaths could be prevented by the provision of high-quality medical interventions termed 'signal functions' for emergency obstetric and newborn care (EmONC) defined by the United Nations agencies [8, 9]. These interventions listed in Table 1 have been identified based on a review of existing literature, the latest WHO guidelines, and their importance in enabling early identification and management of life-threatening complications in both mothers and newborns [10–12].

There are many in-service training packages designed to improve maternal and newborn health in LMIC settings [14]. These training packages tend to cover one or more clinical areas listed in Table 1 and there is also some positive research evidence showing the effectiveness of these training packages in LMIC settings [15–17]. Healthcare workers provide a variety of services across the continuum of care (from pregnancy to postnatal/newborn care) and may benefit more from integrated in-service trainings, i.e., trainings where they are taught comprehensively on multiple topics, for example- routine and emergency obstetric and neonatal care [18]. However, it is likely that greater transfer of knowledge and skills may happen with stand-alone trainings focused on the acquisition of specific clinical skills and learning of specific topics. Integrated trainings are argued to be more cost-effective, reduce absenteeism, cause less disruption of service delivery, and are more efficient since health workers are trained on multiple topics in one training. For example, stand-alone training programs often result in the same health worker being called multiple times from their work location to undergo repeated off-site trainings. On the other hand, integrated training programs have their drawbacks too: their scope is wider (multiple topics are taught) and they are longer compared to vertical trainings, all of which could compromise skill acquisition as well as training quality. On the ground experiences, have also shown that with integrated trainings, there can be a tendency to minimise or omit certain topics depending on the trainer's expertise and interest. The evidence base on whether healthcare workers tend to benefit more from stand-alone trainings compared to integrated trainings in maternal and newborn health (MNH) is limited [19].

Broader questions remain about the overall effectiveness of any type of training programs, with a recent systematic review concluding that there is a need to evaluate the effectiveness of educational interventions on health worker performances and patient outcomes [20]. Training

**Table 1. Essential maternal and newborn care interventions for routine care, basic and comprehensive emergency obstetric and newborn care [10, 11, 13].**

| Dimensions of facility care | Obstetric services | Newborn services |
|---|---|---|
| **1. Routine care (for all mothers and babies)** | Monitoring and management of labor using partograph | Thermal protection |
| | Maternal infection prevention measures (handwashing, gloves) | Immediate and exclusive breastfeeding |
| | Active management of the third stage of labor (AMTSL) | Neonatal Infection prevention including hygienic cord care |
| | | Preparedness for neonatal resuscitation |
| **2. Basic emergency care** | Administration of parenteral magnesium sulphate for pre/ eclampsia | Administering antibiotics for preterm or prolonged rupture of membranes (P/PROM) to prevent infection |
| | Performing assisted vaginal delivery | Administration of Corticosteroids in preterm labor |
| | Administration of parenteral antibiotics for maternal infection | Resuscitation with bag and mask of non-breathing baby |
| | Administration of parenteral oxytocin in the third stage of labor | Kangaroo Mothers Care (KMC) for premature/very small babies |
| | Manual removal of retained placenta | Alternative feeding techniques if baby unable to breastfeed |
| | Removal of retained products of conception | Administration of injectable antibiotics for neonatal sepsis |
| | | Prevention of Maternal to Child Transmission (PMTCT) if the mother is HIV-positive |
| **3. Comprehensive emergency care** | Basic emergency care plus surgery (e.g. cesarean) and blood transfusion | Intravenous fluids |
| | | Safe Oxygen–Bubble continuous positive airway pressure (bCPAP) |

programs are often time-intensive and may have limited impact if newly trained health workers are unable to apply these skills and knowledge while providing clinical services [10, 21]. Training programs may also have limited impact due to various other factors such as poor design or suboptimal delivery; lack of necessary equipment, supplies, and infrastructure; poor organization and management at facilities; frequent staff turnover; frequent rotation of staff; lack of post-training support; or lack of supportive supervision and ongoing mentoring [20, 22, 23]. Generally, information on such determinants, particularly facility environments, would be beneficial for planners to understand how, when and where learners will have to apply their newly acquired knowledge and skills [11], but such contextual information is often not considered while planning.

With this background, we chose to review the existing MNCH training packages in Nepal and Ethiopia, two LMICs from Asia and Africa to help collate the packages and the implementation experiences from key health workers in these two countries. Ethiopia's maternal mortality ratio (MMR) was 353 per 100,000 live births, and 28% of births were attended by skilled health personnel in 2015 [24, 25]. The neonatal mortality rate (NMR) reduced from 37 per 1000 live births [26] in 2011 to 29 per 1000 live births in 2016 [24]. Encouragingly, the number of women coming to deliver at health institutions increased to 26.2% in 2016 [27]. The targets for 2030 are to reduce the MMR to 70 per 100,000 live births, NMR to 12 per 1000 live births and improve coverage of births attended by skilled health personnel to 90% [28, 29]. In Nepal, 58% of births were attended by skilled birth attendants (SBA) and home deliveries remained high at 43% in 2016 [30]. The MMR was 259 per 100,000 live births [31]. Neonatal and infant mortality rates are 21 and 32 per 1,000 live births respectively [30]. The Government of Nepal (GON) aspires to reduce the MMR to 112 per 100, 000 live births and NMR to 13 per 1,000 live births by 2030 [32].

These countries were selected since they had active Maternal Child Survival Program (MCSP) activities focussed on maternal and newborn health. There was a high level of interest in conducting the study from both countries and national staff were available to facilitate local data collection efforts. Both countries have made considerable progress in improving maternal, newborn and child health indicators over the past two decades.

This study aims to describe the differences in the training content of existing, government-approved MNH training packages and capture implementation experiences from key stakeholders regarding the implementation of these training packages in Ethiopia and Nepal. We validated our findings with national experts and stakeholders in both countries and jointly developed recommendations for strengthening in-service trainings for maternal and newborn health in Ethiopia and Nepal. We refrained from making cross-country comparisons and focused, rather, on describing the strengths and weaknesses of training content and implementation in each country separately.

## Materials and methods

### Study design

This is a mixed-methods study using a quantitative analysis of technical contents in training materials used in Ethiopia and Nepal, supported by a qualitative component comprising of key informant interviews to better understand the implementation approaches utilized by various training initiatives [19, 33]. The study was conducted in Ethiopia and Nepal between August 2018 and March 2019.

### Study methods

Before the start of the assessment, we obtained approvals from each country's Ministry of Health (MoH). Independent national consultants (with extensive experience as MNH trainers) collaborated with the MoH to identify relevant training packages targeted towards doctors, nurses and midwives based upon an agreed selection criterion.

Available training materials were identified by the senior in-country study coordinator together with training focal points of the MoH in both countries. Inclusion criteria included MNH training packages developed after the year 2000 and which focused on skilled birth attendants (doctors, nurses and midwives). Exclusion criteria included all training packages with materials that were not officially endorsed by the MoH, did not have a specific focus on doctors, nurses and midwives working in health facilities, or if the learning materials that were incomplete such as stand-alone job aids, policy guides, program manager guides, materials on quality improvement alone. We assessed 12 training packages in Ethiopia and 15 in Nepal for eligibility. After exclusion based on our criteria, we had a final selection of 7 training packages from Ethiopia and 9 from Nepal included in this study (Fig 1). The training packages analysed in this paper included both stand-alone and integrated packages.

We validated our quantitative findings with local experts through a series of key informant interviews and validation workshops to determine that what we identified were true gaps and not deliberate omissions due to contextual factors.

For the qualitative assessment, we conducted face-to-face semi-structured interviews with 12 key informants in Ethiopia and 16 in Nepal. The purpose of the qualitative interviews was to generate insights into implementation approaches utilised by various training packages. We used a purposive sampling technique to identify relevant stakeholders that were involved in organizing and facilitating MNH training packages. We took detailed interview notes and audiotape recordings were also made for future reference. Interviews were conducted until saturation was reached.

### Data collection

Training packages were reviewed by two independent researchers (Ethiopia- MS and GS; Nepal-BB and NK) using the quantitative assessment tool or the data extraction template (S1

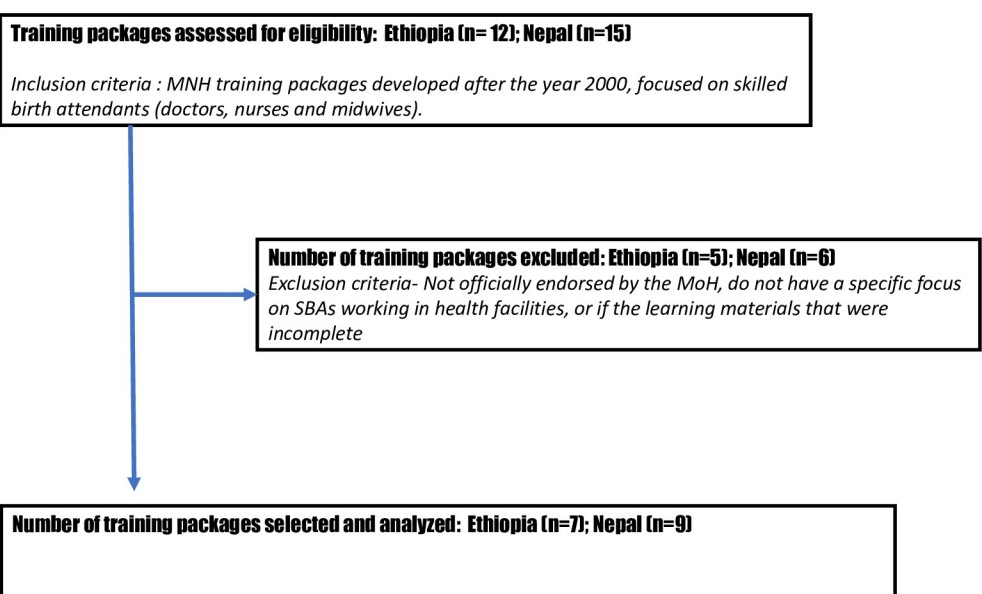

**Fig 1. Flowchart showing steps for auditing the training materials in Ethiopia and Nepal.**

Questionnaire). This was used to identify the presence or absence of essential interventions in each training package. An excel sheet was used to enter and summarize binary responses (presence = Yes or absence = No).

National consultants also helped to identify participants for the key informant interviews. Participants were purposively selected and included MoH technical focal persons for maternal and newborn health; training focal points, representatives of partner organizations supporting MNH training packages; facilitators and learners that received either the stand-alone or the integrated training activities. Most respondents had a medical and public health background.

## Study tools

For the quantitative assessment, a comprehensive data extraction template was developed (available as S1 Questionnaire) which captured information on various training elements such as the type of learning activities, trainer profile, participant/trainer ratio, methodologies to evaluate competencies, time allotted for practical sessions and clinical exposure, as well as technical content for routine, basic and comprehensive emergency obstetric and newborn care. The data extraction tool was based on validated WHO guidelines [12] and was developed jointly with global maternal and newborn experts based on our framework presented in Table 1. The extraction tool gave equal weight to all interventions since there is no scientific basis for giving intervention specific weights and we wanted to be transparent. A semi-structured interview schedule was developed for the key-informant interviews. The interview guide is available as a (S1 File).

## Data analysis

For the quantitative analysis, we collected data on all variables for routine care, basic and comprehensive emergency obstetric and neonatal care that are outlined in Table 1. Variables under different technical areas were coded as '1' if available or '0' if not available in different training materials. All data were entered and analysed in Microsoft Excel. Frequencies were computed for all variables and data entered was cross-checked with original forms. After

cross-checking for accuracy and completeness, summary scores were calculated for each clinical practice. Proportions were generated for each clinical practice which was defined as the total number of 'yes' responses divided by the total number of interventions in that clinical practice. As an example, a proportion of 50% implies that the training package contained 50% of the recommended interventions for that clinical practice.

The key informant interviews were conducted in Amharic and Nepali. The findings were transcribed in English and analyzed using Microsoft Excel. All the interviewers were involved in the transcription. A thematic analysis approach was utilized. To ensure consistency of the data, two researchers (MS and GS—Ethiopia and BB and NK- Nepal) independently reviewed responses and agreed on a set of codes. A codebook was developed to define the codes. Inter coder reliability between two coders was assessed manually using Microsoft excel Themes such as challenges for scaling up MNH training packages, national databases for training, and potential solutions and innovations were captured.

The mixed-methods approach allowed us to identify gaps in the technical content for various clinical interventions (quantitative analysis) and helped us generate insights into the context and weaknesses in implementation approaches (qualitative analysis). Preliminary findings from the audit of training packages and the qualitative interviews were presented at workshops in Ethiopia and Nepal where findings were validated with the insights of national experts working in maternal and newborn health in both countries.

## Ethics and consent to participate

Ethical approval was obtained from the Save the Children's Ethics Review Committee. United States Agency for International Development (USAID) reviewed and contributed to the development of the study protocol. Approvals were sought from the Ministries of Health in both countries before undertaking data collection. The research involved the desk review of training materials and interviews to capture respondents' opinions related to MNH training packages in Ethiopia and Nepal. The study did not test interventions or collect biological samples. Therefore, there was no direct risk associated with this study. Data collectors obtained written informed consent from participants before each interview. Before the interview, all participants were informed about the study, its sponsorship, confidentiality of any data collected and their ability to stop the interview at any time they desired.

## Results

We analysed 7 MNH training packages in Ethiopia and 9 packages in Nepal. In Ethiopia, training packages ranged from short (3 days) vertical training packages focused on essential care for every baby (ECEB) and Prevention of mother to child transmission (PMTCT) to three-month-long training packages on comprehensive emergency care. Similarly, in Nepal, training packages ranged from short training packages that were delivered over one day (Helping babies breathe) to longer training packages such as the SBA training (60 days) and Advanced SBA training package (70 days). Table 2 provides further details on the duration of training packages and cadres eligible to receive these training packages in Ethiopia and Nepal.

### Newborn care interventions in routine care

In Ethiopia, neonatal resuscitation was addressed comprehensively in all materials except in the IMNCI manuals (89%) in terms of components related to routine essential newborn care. Newborn infection prevention practices including hygienic cord care were found to be incomplete in BEmONC (70%), CEmONC (70%), and IMNCI (80%) manuals. The BEmONC and CEmONC manuals had not incorporated newer recommendations such as delayed cord

**Table 2. Summary of maternal and newborn health trainings in Ethiopia and Nepal.**

| Country/ Training package | Duration of the training | | | Cadre eligible for trainings | | | | | Integrated (YES/NO) |
|---|---|---|---|---|---|---|---|---|---|
| | Theory (days) | Practical (days) | Total duration (days) | Doctor-MBBS/MD | Doctors—Specialists | Nurses (certificate, BA, MSc) | Midwife / ANM | Others–IESO | |
| *Ethiopia* | | | | | | | | | |
| Essential Care for Every Baby (ECEB) | 0 | 3 | 3 | Yes | No | Yes | Yes | No | NO |
| Integrated Management of Neonatal and Childhood Illnesses (IMNCI) | 6 | 1* | 6 | Yes | No | Yes | No | No | NO |
| Neonatal Intensive Care Unit (NICU) | 10 | 18 | 28 | Yes | Yes | Yes | Yes | No | NO |
| Prevention of Mother to Child Transmission (PMTCT) | 3 | | 3 | Yes | No | Yes | Yes | Yes | YES |
| Basic Emergency Obstetric and Newborn Care (BEmONC) | 8 | 10 | 21 | Yes | Yes | Yes | Yes | Yes | YES |
| Comprehensive Emergency Obstetric and Newborn Care (CEmONC) | 26 | 64 | 90 | Yes | Yes | Yes | No | Yes | YES |
| Newborn Care (NBC) | | 3 | 3 | Yes | No | Yes | Yes | No | NO |
| *Nepal* | | | | | | | | | |
| Skilled Birth Attendant (SBA) | 12 | 48 | 60 | Yes | Yes | Yes | Yes | NA | YES |
| Advanced Skilled Birth Attendant (ASBA) | 7 | 63 | 70 | Yes | Yes | No | No | NA | YES |
| Postnatal care (PNC) | 10 | 17 | 27 | No | No | Yes | Yes | NA | YES |
| Comprehensive newborn care Level 2 (CNBC L2) for Nurses | 8 | 8 | 16 | No | No | Yes | Yes | NA | NO |
| Comprehensive newborn care Level 2 (CNBC L2) for Doctors | 4 | 2 | 6 | Yes | Yes | No | No | NA | NO |
| Facility-based Integrated Management of Neonatal and Childhood Illnesses (FB-IMNCI) | 4 | 2 | 6 | Yes | Yes | No | No | NA | NO |
| Maternal and Newborn Health (MNH) update | 2 | 1 | 3 | Yes | No | Yes | Yes | NA | YES |
| Helping Babies Breathe- Version-2 (HBB2) | | 1 | 1 | Yes | Yes | Yes | Yes | NA | NO |
| Prevention of Mother to Child Transmission (PMTCT) | 5 | | 5 | Yes | Yes | Yes | No | NA | NO |

* In IMNCI there are clinical sessions on days 3, 4 and 5. Each session lasts about 2:30 hours.

clamping. Only two manuals (ECEB and NICU) covered basic newborn care interventions comprehensively.

In Nepal, for components related to basic newborn care; thermal protection was incomplete in SBA (86%), ASBA manuals (43%), FB-IMNCI manual (86%), clinical mentor guide and MNH updates package (43%), HBB-version 2 manual (71%) and PMTCT manuals (29%). Immediate and exclusive breastfeeding was found to be incomplete in the ASBA (63%), FB-IMNCI (88%), MNH update, HBB-2 (50%) and PMTCT manuals (29%). Similarly, neonatal infection prevention including hygienic cord care was found to be incomplete in all manuals except the SBA manual. Preparedness for neonatal resuscitation was found to be incomplete in FB-IMNCI (89%) manuals and absent from the PMTCT manual.

## Newborn care interventions in basic emergency care

In Ethiopia, antibiotics for preterm premature rupture of the membranes (P/PROM) to prevent infection was covered only in BEmONC and CEmONC manuals. Antenatal corticosteroids for preterm labor was covered well in BEmONC and CEmONC manuals (90%). Neonatal resuscitation with bag and mask in case of a non-breathing baby was covered well in all manuals except the NICU and PMTCT manuals. KMC technical content was found to be incomplete in IMNCI (33%) and completely absent in NICU and PMTCT materials. Injectable antibiotics for neonatal sepsis were absent in NICU and PMTCT training materials. Care for HIV infected newborns was covered comprehensively in PMTCT and IMNCI manuals but absent from all the other manuals in Ethiopia. The BEmONC manual in Ethiopia did not recommend antiretroviral prophylaxis or refer participants to relevant sections of the national guidelines.

In Nepal, for basic emergency care interventions, antibiotics for P/PROM to prevent infection were fully covered (100%) in the SBA and ASBA manuals and were absent from all other training materials. Resuscitation with bag and mask of the non-breathing baby was covered comprehensively (100%) in all training materials except the PMTCT materials. None of the training materials covered antenatal corticosteroids for preterm labor since corticosteroids are still not included in the national standards. Kangaroo mother care for premature or very small babies was not covered in clinical mentors' guide, HBB- version 2 and PMTCT materials. Management of the HIV-exposed infant was covered to varying degrees in different training materials.

## Newborn care interventions in comprehensive emergency care

In Ethiopia, for comprehensive emergency care interventions, fluid management in newborns, safe oxygen therapy and b-CPAP therapy were covered comprehensively (100%) in the NICU training materials, but these interventions were missing in the remaining six manuals. The CEmONC manual did not comprehensively cover newborn resuscitation, stabilisation, initiating effective ventilation, preventing hypothermia and hypoglycaemia. The manual also did not provide instructions for referral to a higher centre. The comprehensive emergency care interventions were also not linked to the relevant sections of the NICU manual.

In Nepal, for comprehensive emergency care interventions, fluid management in the newborn was covered comprehensively (100%) in CNBC level– 2 for nurses and doctors but b-CPAP (100%) and safe oxygen therapy (100%) were only covered in the CNBC level- 2 materials for doctors. None of the other materials covered these newborn comprehensive emergency care interventions. Fig 2 summarises our findings related to newborn care interventions in Ethiopia and Nepal.

## Routine maternal health interventions at the time of birth

In Ethiopia, both the BEmONC and CEmONC manuals appear complete (100%) but none of the other newborn health-focused manuals covers routine maternal health interventions. The PMTCT manual covers monitoring of labor using a partograph and infection prevention measures but does not cover active management of the third stage of labor.

In Nepal, labor monitoring using partograph was covered in the SBA manual (93%), ASBA manual (100%), MNH updates (87%) but not covered in any of the other training materials. Infection prevention measures were covered comprehensively in SBA, ASBA materials and CNBC for level– 2 nurses but were incomplete in HBB- 2 (80%) and PMTCT (80%) and absent in PNC, MNH updates, CNBC- nurses and facility-based IMNCI. Active management of the

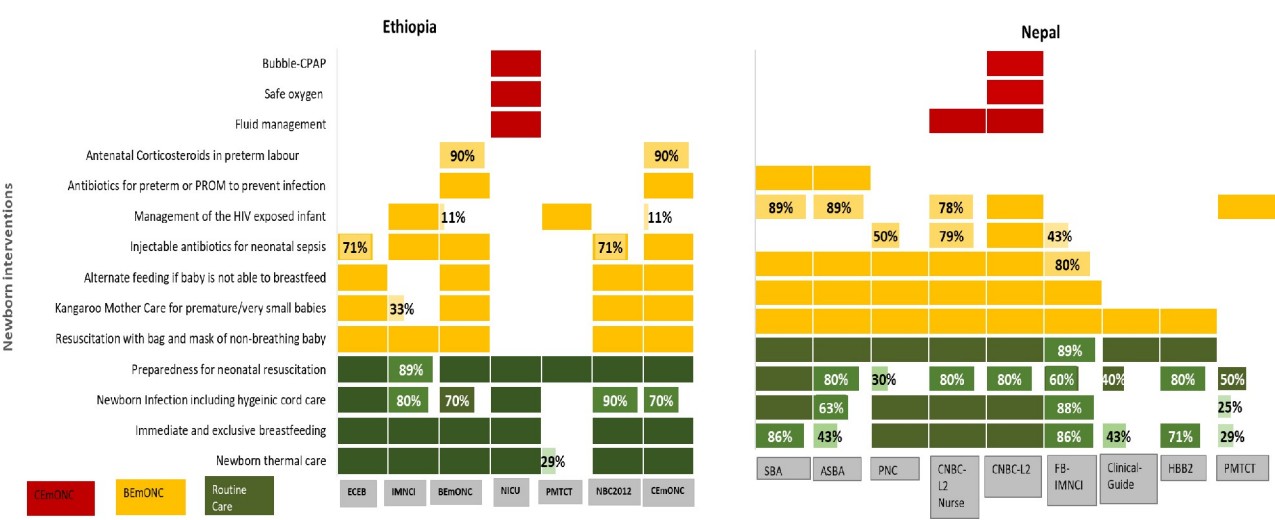

**Fig 2. Newborn care interventions in routine, basic and comprehensive emergency care in Ethiopia and Nepal.**

third stage of labor was also covered to some extent in the SBA (89%), ABSA (67%) and MNH updates (33%) but missing in all the other manuals that were reviewed.

## Maternal health interventions in basic emergency care

In Ethiopia, the national BEmONC manual was complete with all clinical interventions duly reflected (100%). However, the CEmONC manual did not cover certain details on prophylactic antibiotics before caesarean sections for the prevention of maternal infections. None of the other newborn health focussed manuals discussed maternal health interventions in basic emergency care.

In Nepal, for maternal health basic emergency care interventions, all signal functions were covered adequately in SBA and ASBA materials except parenteral antibiotics for maternal infections, which was covered up to 89% in the SBA and 78% in the ASBA manual. The MNH update manual only focused on parenteral magnesium sulphate, assisted vaginal delivery and parenteral oxytocic drugs for hemorrhage, and did not cover other signal functions. None of the other newborn health manuals covered maternal signal functions in basic emergency care interventions.

## Maternal health interventions in comprehensive emergency care

In Ethiopia, the CEmONC manual was found to cover a majority of technical contents (71%) but finer details such as what precautions should be taken during caesarean section, what are the complications following caesarean section and management of complications such as modified b-lynch sutures and obstetric hysterectomy were absent in other training materials. The regimen for prophylactic antibiotics before Caesarean section was also absent in the CEmONC manuals.

In Nepal, the ASBA manual covered all aspects of the cesarean section whereas the SBA manuals covered 57% and the MNH update covered about 50% of the content. These manuals were found to cover indications for caesarean and when to refer for complications of pregnancy but missed other details. Fig 3 below summarizes maternal care interventions in Ethiopia and Nepal.

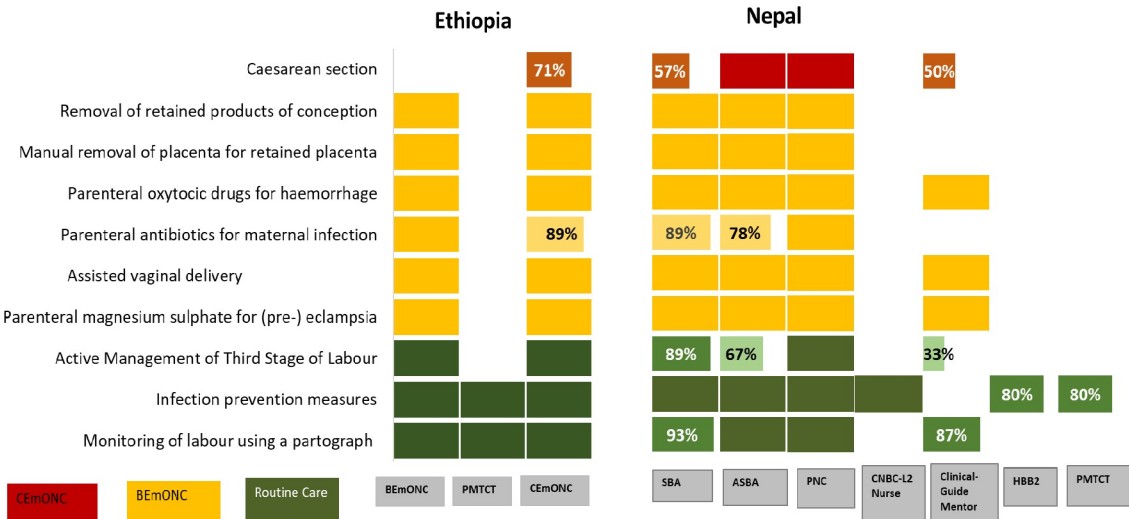

**Fig 3. Maternal care interventions in routine, basic and comprehensive emergency care in Ethiopia and Nepal.** We removed ECEB, IMNCI NICU, NBC-2012 manuals (Ethiopia) and CNBC- Level 2 for doctors, FB-IMNCI for doctors (Nepal) from the graph since they did not have any maternal health components.

## Summary of qualitative findings

The qualitative interviews with the key informants supplemented the quantitative findings by giving a better understanding of implementation approaches and experiences of the stakeholders with the training packages in Ethiopia and Nepal. The qualitative data is organized into themes that are broadly related to the technical content of training materials and implementation approaches (before, during and after training). Specifically, themes were related to planning, quality, technical content, scaling-up, post-training skills retention, training-related metrics and training management issues. Key themes that emerged from the key informant interviews are summarized in Table 3.

**Training planning.** Most participants expressed that in cases where the same health worker provides MNH services and when appropriate, integration may be a cost-effective option. Some participants suggested that a promising alternative strategy would be to first measure the existing quality of care provided by health workers and then design or implement specific technical modules based on the deficiencies identified from such an assessment rather than taking a universal approach towards training health workers. The respondents reported that implementing such a strategy where specific modules (or trainings) are implemented to address identified gaps in existing quality of care will help to improve the knowledge and skills of health workers.

Another recommendation by the participants was around strengthening the pre-service curriculum for MNH. The participants reported that since the design and development of a pre-service curriculum is a time-consuming and challenging process, it tends to remain unchanged for many years. However, strengthening areas that are weaker or outdated has the potential to bring about large-scale changes in countries. Another planning issue highlighted by participants was that suitable participants that fulfil the selection criteria are not always invited to attend the trainings. It was emphasised by the participants that training health workers that have no role in providing MNH services, is a waste of resources and a significant opportunity cost. Training information systems was also identified as a major planning challenge by participants in both countries. For example, information on which health worker has

**Table 3. Summary of qualitative findings.**

| Themes | Summary of key findings from the interviews. |
|---|---|
| Training planning | • The actual integration of technical functions at the level of the peripheral health worker is not considered while planning |
| | • Suitable participants that fulfill the selection criteria are not always selected for trainings |
| | • Limited numbers of certified neonatal care training centers limit expansion efforts |
| | • Trainings should be conducted in response to performance assessments or measurement of existing service quality |
| | • Strengthening pre-service curricula holds promise but is a neglected area. |
| | • Training information systems are not functional in both countries |
| Training quality | • Ensure quality during the rollout of the trainings |
| | • Lack of newborn cases for practical sessions |
| | • Practical sessions are not taken seriously by both the trainers and trainees |
| | • Facilitators may or may not have received training from the trainers. |
| | • There is usually an inadequate number of facilitators. |
| | • NICU trainings are resource-intensive but do not employ competency-based training methods |
| | • *Dilution effect-* new topics are constantly added to trainings without thinking about quality |
| Limited technical content for newborn health | • Core skills necessary for providing newborn care at primary, secondary and tertiary levels |
| | • MNH training materials do not cover newborn health comprehensively and some are outdated |
| | • Some training materials (IMNCI and PMTCT) are not focused on skill acquisition |
| | • Trainers' expertise and familiarity with certain technical topics mean that all topics do not receive the same level of attention or enthusiasm |
| Scaling up training packages | • Cost-effectiveness to scale up long and expensive trainings |
| | • Rapid scale-up of trainings can hamper the quality of learning |
| | • Limited numbers of expert newborn health trainers |
| | • Managing attrition of trained human resources or constant turnover of skilled staff |
| | • There are many stand-alone training initiatives and duplication of efforts. |
| Post-training skill retention and application of skills | • Newly learnt skills can deteriorate or be lost if there are not enough opportunities to practice |
| | • Skills retention by learners in high caseload and low caseload clinical sites |
| | • Linking training and retraining to continuing professional development |
| | • MoH staff are not always able to supervise and follow up after training |
| | • There are no resources allocated for post-training follow up |
| Training metrics | • Use of pre-test and post-test scores–adequate representation of all technical areas |
| | • Better use of pre and post-test scores |
| | • Monitoring learners' progress in service delivery after they leave the trainings |
| Training management | • Lack of budgets affects training preparations |
| | • If there are no per diems, learners don't seem as motivated |
| | • Lack of equipment and supplies for clinical practice such as Ambu bags and radiant warmers affects training quality. Even when supplies are available, they may be poorly maintained |
| | • Some integrated trainings are of long duration, therefore, resource-intensive (BEmONC, CEmONC and NICU) and interrupt service delivery |
| | • There is a lack of adequate cases for the practical sessions |
| | • Overcrowding at training sites can be problematic since these training sites are also used by other students (nurses, doctors) |

received training or where they are posted are hard to obtain. Participants suggested that there needs to be a greater investment in developing or strengthening functional and usable health training information system which can support planning efforts.

**Training quality.** Ensuring high-quality trainings are important, particularly as the training cascades down to peripheral levels. Participants reported that despite the Ministry of Health investing significant resources into preparing clinicians as master trainers, trainings are not a part of the official job description and hence trainers are often reluctant to go for trainings in peripheral areas. Participants from both countries highlighted the need to thoughtfully select skilled trainers who are committed and invest in creating an enabling environment for them with appropriate incentives so that they are retained within the system and training quality is maintained as trainings are expanded. Other quality-related challenges reported by the participants included a lack of clinical exposure during trainings, inadequate numbers of cases, limited training centres, resource constraints and lack of skilled facilitators (Table 3). Participants also highlighted that a '*dilution effect*' may occur as a result of integrating various modules within one training package. For example, in Nepal, participants noted that the integration of HBB within SBA modules resulted in reduced training time for other modules and a change in training methodology. Another example given was that after the integration of KMC into SBA trainings, binding the baby to the mother received attention but other components of KMC did not receive adequate attention.

**Limited technical content for newborn health.** It was reported that there is a need to update existing training materials to reflect recent advances in global guidelines and ensure a focus on skills transfer and competency-based training methods. In certain cases, integration of technical content has also led to confusion amongst learners on practical issues. One example from the participants was about the difficulty in knowing the sequence for providing oxytocin injection for AMTSL when a non-breathing newborn also required resuscitation or the right sequence for applying chlorhexidine to the cut cord and initiating immediate skin to skin contact. Participants also reported that trainers' expertise and preferences often result in some sessions receiving more attention than others. For example, newborn health tends to receive less importance if an obstetrician conducts the training and vice versa. Lastly, participants highlighted the need to define core competencies necessary for providing newborn care at primary, secondary and tertiary levels.

**Scaling up training packages.** Issues discussed under the theme of scaling-up trainings focused on high costs associated with long duration of trainings, attrition and turnover of staff, and challenges of sustaining quality of trainings at scale.

One participant from Nepal highlighted that, "*although approximately 7,000 SBAs were trained on the SBA package over the past decade, less than half of those trained remain in the public sector. Participants also stated that rapidly scaling up trainings to meet coverage targets without adequate attention to training quality does not lead to the desired impact.*"

One participant from Ethiopia also mentioned that "*Often, there are numerous vertical, or donor led initiatives that contribute to duplication of efforts and do not strengthen existing national systems.*"

**Post-training skill retention and application of skills.** The retention and application of newly acquired skills in routine clinical practice was an important theme that emerged from the key informant interviews. Participants reported that skills deteriorate in circumstances where health workers do not have the opportunity to practice them. Respondents from both countries reported that, although mentoring and supportive supervision were recognised as important strategies, there were many associated challenges. For example, mentors need to be released from their daily clinical duties, a replacement must be found so that services are uninterrupted, and incentives for mentors must be agreed upon and obtained. Participants from

both countries suggested that implementation research projects are necessary to answer questions around effective strategies for mentorship and supervision.

One participant said, "*we need to learn more about effective models for mentorship and supervision—who should follow up, how frequently, how much does it cost, what is the process, where (on or off-site), what works and what does not work in a particular context*".

**Training metrics.**   Respondents in both countries suggested that trainings must do better in terms of assessing learners' progress during training and after they go back to work. Although pre and post-test scores are routinely utilised as most trainings follow competency-based learning methodologies, these scores tend to be used only for certification purposes. Participants emphasised that there are many opportunities to use them systematically such as linking them to continuing professional development and professional licensing.

**Training management.**   Participants reported a lack of adequate budgets and logistical challenges as consistent realities especially when trainings are implemented by the Ministry of Health. Participants further highlighted that the resource constraints often mean that obtaining adequate training material and supplies and recruitment of expert trainers is problematic. Further, if programs cannot provide daily subsistence allowances, learners are not as motivated. Other issues reported included an inadequate number of cases for the clinical sessions, limited numbers of skilled trainers, and overcrowding at training sites. Lastly, participants highlighted that training management guidelines are often neglected and must be followed diligently so that quality can be maintained during implementation.

## Discussion

This mixed-methods study uncovered several gaps in the training curricula for routine care, basic and comprehensive emergency obstetric and newborn care in Ethiopia and Nepal. Key informant interviews provided additional insights and generated useful recommendations for strengthening training programs and approaches in both countries. Overall, we found significant gaps in technical content for newborn health in Ethiopia and Nepal. Areas found to be weak for routine care of newborns included preparedness for neonatal resuscitation, care for the small baby at home and newborn infection prevention including hygienic cord care.

Although neonatal resuscitation is one of the most urgent clinical situations in pediatrics, it was found to be missing in IMNCI guidelines in Nepal. It would be beneficial for countries to close down these gaps in line with the WHO standards for improving the quality of maternal and newborn care in health facilities [12]. For newborn interventions in basic emergency care, prophylactic antibiotics for P/PROM, antenatal corticosteroids for preterm labor, injectable antibiotics for sepsis and PMTCT were found to be incomplete. In Nepal, none of the training materials covered antenatal corticosteroids for preterm labor since they are not recommended for use by personnel other than qualified physicians. WHO guidelines (2015) currently recommend antenatal corticosteroids for women at risk of preterm birth from 24 to 34 weeks of gestation when gestational age assessment can be accurately undertaken, preterm birth is imminent, there is no maternal infection, and adequate facilities for the management of preterm birth are available at the secondary or tertiary level [34]. PMTCT appears to be a stand-alone entity in both Ethiopia and Nepal and does not incorporate most MNH components. Since mother to child transmission accounts for 90% of HIV infections in children, all health workers must be informed about PMTCT guidelines [35]. Further, babies born to HIV positive mothers tend to be preterm [36] and will need additional feeding and thermal care support [37], hence PMTCT manuals should incorporate home-based care of small babies. Similarly, all MNH trainings should cover the management of the HIV exposed infant [38]. As the prevalence of preterm births in Ethiopia (10.5%) and Nepal (9.3%) is high, this finding is particularly

relevant and trainings should include special care for preterm births [39, 40]. Newborn interventions in CEmONC were found to be generally weak in both countries and need additional strengthening.

Not surprisingly, maternal health interventions were generally missing from newborn health training materials in both countries. AMTSL was found to be weak in Nepal in terms of routine maternal care interventions. For maternal health interventions in basic emergency care, all signal functions except parenteral antibiotics for maternal infections were covered adequately. For maternal health interventions in comprehensive emergency care, the advanced obstetric training materials covered all topics in detail except the management of complications following caesarean sections. These are all important signal functions for high-quality basic and emergency care services [11].

Results from our quantitative and qualitative analysis suggest four key avenues for improving training in Ethiopia and Nepal. First, there is a need to strengthen MNH technical content, improve alignment between training packages, and think carefully about the design and delivery of future trainings. Our findings are in line with previous studies examining SBA trainings in LMICs, which reported that education and training for SBAs greatly varied between countries in terms of duration and contents of the training [41]. Researchers found wide variation in the skills and competencies of staff across countries in terms of their ability to manage routine and emergency conditions [41]. Gaps in technical content identified through this rapid assessment have already been shared with the Ministries of Health and national experts in both countries.

Second, alternative approaches for training health workers should be explored. Designing and implementing specific training modules in response to deficiencies identified from the measurement of QoC could be a promising strategy to improve clinical quality. Clinical practice observations have been utilized with success in many settings to assess QoC [42]. Based on the findings of such observations, health workers should be provided with opportunities to take blended, self-directed, modules on individual topics so that any gaps in knowledge and skills can be closed down. Evidence from Tanzania suggests that remote or blended learning approaches could be feasible in low resource settings [43]. Innovative blended approaches that require learners to complete some preliminary reading and assignments before they come for trainings is a way to shorten overall training duration. Perhaps the best way forward is to use a variety of complementary approaches starting with high-quality pre-service trainings [19].

Third, there is a need to focus on skill retention after training, improve ongoing mentoring and identify better ways to provide a supportive environment for health workers to apply their newly learnt knowledge and skills. Training transfer is linked to work environments and staff's perception of their work environment [44, 45], therefore, it is important to strengthen overall health systems. Program evidence suggests that health workers need repeated opportunities for training and that mastery of skills requires repeated practice. In Nepal, program reviews have found that skills deteriorate rapidly if health workers do not have opportunities to practice their newly acquired skills [46]. Also, skill retention is likely to vary depending on the work setting, opportunities to practice [47] and other factors such as being based at a high- or low-volume site, being based at a primary, secondary or tertiary level health facility, availability of instruments, essential commodities and supplies, and support received from facility management and leadership. However, there is limited evidence on the impact of in-service trainings on actual clinical outcomes in LMICs [19]. Hence, future studies that are well-designed and examine the actual impact of trainings on clinical practices and patient outcomes are urgently needed.

This study adds to the limited but growing evidence-base on the content of various in-service training materials and their implementation experience in both countries. Finally, our study indicates a need for greater investments in developing and strengthening functional

training management information system for keeping track of trainings received by health workers which can better support planning efforts in both countries [48]. Further research on effective training management information systems is urgently needed. Opportunities to introduce and institutionalize platforms for continuous professional development by professional councils could also be pursued in both countries [49].

## Limitations

Our analysis is subject to several limitations. First, we did not develop or test any hypothesis in our analysis as our aims were primarily descriptive. Second, we chose training materials and key informants in a purposive manner, which may limit the generalizability of our results to some extent. Third, although all key informant interviews were semi-structured and conducted by experienced interviewers, it is possible that interviewer bias may have influenced some of the comments recorded or question is asked. Fourth, we were unable to observe actual training sessions or measure the existing quality of care provided by health workers who had received any of the trainings we reviewed. Finally, we excluded training materials targeted at community-based health workers–a key part of the MNH service delivery system in both Ethiopia and Nepal–because our review focused specifically on materials for training clinically qualified health workers. Future studies may want to investigate the content and quality of MNH trainings provided to community-based health workers.

## Conclusion

We found several gaps in the technical contents of the maternal and newborn health training curricula in Ethiopia and Nepal. The existing training packages could be improved by strengthening the missing technical content, improving alignment between different MNH training packages, using innovative methods to redesign existing training packages, better supporting health workers in terms of skill retention, and developing training information systems to keep up-to-date records on trainings received by health workers. These findings and recommendations may be of interest to other LMICs facing similar challenges in training content development and delivery.

## Supporting information

**S1 Questionnaire. Training curriculum data extraction tool.**
(DOCX)

**S1 File. Interview guide.**
(DOCX)

**S2 File. Key informant consent form for interviews.**
(DOCX)

**S3 File. Information sheet for key informants.**
(DOCX)

**S1 Table. Data file.**
(XLSX)

## Author Contributions

**Conceptualization:** Gaurav Sharma, Yordanos B. Molla, Adhish Dhungana, Abeba Bekele, Neena Khadka.

**Data curation:** Gaurav Sharma, Yordanos B. Molla, Bindu Bajracharya, Deepak Jha, Abeba Bekele.

**Formal analysis:** Gaurav Sharma, Yordanos B. Molla, Adhish Dhungana, Bindu Bajracharya, Goitam G. Mebrahtu, Shilu Adhikari, Deepak Jha, Yunis Mussema, Abeba Bekele, Neena Khadka.

**Funding acquisition:** Neena Khadka.

**Investigation:** Gaurav Sharma, Yordanos B. Molla, Million Shibeshi, Adhish Dhungana, Bindu Bajracharya, Goitam G. Mebrahtu, Shilu Adhikari, Deepak Jha, Yunis Mussema, Abeba Bekele, Neena Khadka.

**Methodology:** Gaurav Sharma, Yordanos B. Molla, Abraham Tariku, Adhish Dhungana, Shilu Adhikari, Abeba Bekele, Neena Khadka.

**Project administration:** Yordanos B. Molla, Abraham Tariku, Adhish Dhungana, Bindu Bajracharya, Abeba Bekele, Neena Khadka.

**Resources:** Yordanos B. Molla.

**Software:** Gaurav Sharma.

**Supervision:** Gaurav Sharma, Yordanos B. Molla, Million Shibeshi, Adhish Dhungana, Bindu Bajracharya, Abeba Bekele, Neena Khadka.

**Validation:** Gaurav Sharma, Yordanos B. Molla, Million Shibeshi, Abraham Tariku, Adhish Dhungana, Bindu Bajracharya, Goitam G. Mebrahtu.

**Visualization:** Gaurav Sharma, Yordanos B. Molla.

**Writing – original draft:** Gaurav Sharma, Shyam Sundar Budhathoki.

**Writing – review & editing:** Gaurav Sharma, Yordanos B. Molla, Shyam Sundar Budhathoki, Million Shibeshi, Abraham Tariku, Adhish Dhungana, Bindu Bajracharya, Goitam G. Mebrahtu, Shilu Adhikari, Deepak Jha, Yunis Mussema, Abeba Bekele, Neena Khadka.

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
