## [Decision Letter · Decision Letter 0]

3 Feb 2021

PONE-D-20-38462

Analysis of maternal and newborn health training content and approaches to inform future training programs for routine care, basic and comprehensive emergency obstetric and newborn care: Lessons from Ethiopia and Nepal

PLOS ONE

Dear Dr. Sharma,

Thank you for submitting your manuscript to PLOS ONE. After careful consideration, we feel that it has merit but does not fully meet PLOS ONE’s publication criteria as it currently stands. Therefore, we invite you to submit a revised version of the manuscript that addresses the points raised during the review process.

We look forward to receiving your revised manuscript.

Kind regards,

Hannah Tappis, DrPH, MPH

Academic Editor

PLOS ONE

Additional Editor Comments:

This is an interesting article with potential to make a unique contribution to maternal and newborn health policy, programming and research. Please carefully consider feedback from both reviewers, with particular attention to clarification of research questions/objectives and contextualization of findings within the current global evidence base on the topic of focus.

Journal Requirements:

2.) Please include additional information regarding the survey or questionnaire used in the study and ensure that you have provided sufficient details that others could replicate the analyses. For instance, if you developed a questionnaire as part of this study and it is not under a copyright more restrictive than CC-BY, please include a copy, in both the original language as well as the English version already provided, as Supporting Information.

3.) We note that you have indicated that data from this study are available upon request. PLOS only allows data to be available upon request if there are legal or ethical restrictions on sharing data publicly. For information on unacceptable data access restrictions, please see http://journals.plos.org/plosone/s/data-availability#loc-unacceptable-data-access-restrictions.

4.) PLOS requires an ORCID iD for the corresponding author in Editorial Manager on papers submitted after December 6th, 2016. Please ensure that you have an ORCID iD and that it is validated in Editorial Manager. To do this, go to ‘Update my Information’ (in the upper left-hand corner of the main menu), and click on the Fetch/Validate link next to the ORCID field. This will take you to the ORCID site and allow you to create a new iD or authenticate a pre-existing iD in Editorial Manager. Please see the following video for instructions on linking an ORCID iD to your Editorial Manager account: https://www.youtube.com/watch?v=_xcclfuvtxQ

Reviewers' comments:

Reviewer's Responses to Questions

**Comments to the Author**

1. Is the manuscript technically sound, and do the data support the conclusions?

Reviewer #1: Yes

Reviewer #2: Partly

2. Has the statistical analysis been performed appropriately and rigorously? 

Reviewer #1: Yes

Reviewer #2: N/A

3. Have the authors made all data underlying the findings in their manuscript fully available?

Reviewer #1: Yes

Reviewer #2: Yes

4. Is the manuscript presented in an intelligible fashion and written in standard English?

Reviewer #1: Yes

Reviewer #2: Yes

5. Review Comments to the Author

Reviewer #1: General comments: The manuscript analyzes the maternal and newborn health training curricula in Ethiopia and Nepal. The authors conducted mixed qualitative and quantitative methods to assess the maternal and newborn health training content and approaches. The authors concluded that the technical content of training curricula had several gaps that should be addressed and the alignment between different MNH training packages should be improved by using innovative methods.

The manuscript needs improvement in grammar, the writing style and the sentence structure. Copy editing is required.

Below are the specific comments and questions:

Abstract:

In findings, it is strange to separate active management of the third stage of labor from CmONC as it is one of the signal functions. It would be better to include specific gaps in CmONC.

It seems that qualitative findings are missing.

The recommendations need to be adjusted based on the aims and findings of the study. The second and third recommendations seem vague. It would be better to come up with a clear and concise conclusion in the abstract.

Introduction

The introduction section needs a fundamental revision. It is a challenge for a reader to follow the sentences and paragraphs. The logical flow between the sentences and paragraphs needs to be considered.

Also, the definition of the authors for this study should be aligned with evidence and literature. Many statements are not supported by references. The rationale for this study is not well described.

Page 3: It would better to move the table as an annex and briefly describe the table in the introduction.

Page 3, line 6-10: It is hard to understand what the authors are trying to describe here. It does not have a logical flow with the previous sentences.

Page 3, table 1: According to proposed new signal functions by Gabrysch et. al BEmONC should be changed to Basic emergency care and CEmONC to Comprehensive emergency care. Otherwise, it creates confusion among readers.

Page 4, first paragraph: A reference is needed.

Page 4, line 6-7: It could be a perception, may not be supported by evidence.

Page 4: 13-19: Again, it is a challenge to follow. And the references are missing.

Page 5, Second paragraph: The logical flow should be considered.

Page 5: The aims of this study should be clear and easy to understand. It is difficult to grasp the aims of the study.

Page 5: Line 18-21: This sentence needs to be moved to the acknowledgments.

Page 5: The last paragraph should be integrated with the aims of the study followed by possible implications.

Methods:

Again, the logical flow of the ideas should seriously be considered. It’s a challenge for a reader to follow the jumping ideas in these sentences. It would be better to follow a smooth transition from one sentence to another and clarify the linkages.

Page 7: The last two paragraphs are relevant to the introduction section.

The authors need to describe Figure 1. the essential data in the text and clearly explain the inclusion and exclusion criteria.

“Priority was given to in-service training and interventions at health facility setting only.” It is a standalone sentence without any justification or profound description.

The tools development process and justification should further be explained.

It would be better to describe the type of software used for qualitative data analysis and the quality assurance data analysis- intercoder reliability assessment.

Result:

The authors need to consider reorganizing the results section and retain the consistency of presenting the findings. It would be better to include numbers beside the proportion for each clinical practice as number (%).

Page 11: Why did authors present the findings of different packages such as PMTCT, NICU, and IMNCI under the subheading of Newborn care interventions in BEmONC and CEmOC. These are confusing! If the authors are following Sabine Gabrysch et.al proposed framework, then the terms need to be clarified as basic or comprehensive emergency care to avoid confusion.

Presenting the data is not consistent throughout the results section e.g. page 12.

Page 12: Again, “In Ethiopia, for CEmONC, fluid management in a newborn, safe oxygen therapy and b-CPAP therapy were covered comprehensively in the NICU training materials,”. It is confusing to see the CEmONC is mixed up with different training packages. It seems that BEmONC and EmONC packages and signal functions are miss-interpreted in the results section.

Page 13: “the national BEmONC manual was found to be comprehensive”. It seems subjective without presenting the data.

The authors don’t need to present all the details of the findings in the qualitative section. They can focus more on the main and key findings. They also need to include some quotes from the participants of the study to support the findings and make the results more interesting. It would be interesting to see subheadings based on the themes.

Discussion:

The findings which are presented in the results section need to be interpreted in the discussion section but not the other new findings. For example: ”Specifically, newborn infection prevention guidelines were found to be deficient for topics such as hand washing before and after handling babies, the importance of rooming-in with the mother,

benefits of co-bedding, encouragement for early breastfeeding...". It would be strange to see these results for the first time.

It would be better to discuss the qualitative results, not only the recommendations from the participants.

Conclusion:

The first paragraph seems unnecessary. The first sentence of the second paragraph is general. It would be better to make it more specific.

Thank you. Best wishes to the authors.

Reviewer #2: This is an important study with the potential to help us understand gaps in the delivery of maternal and newborn health training content and approaches in Ethiopia, Nepal and more broadly. The paper is well written with only some minor grammatical edits required. The study sought to audit existing training packages in the two contexts of interest, and the authors have effectively defined essential maternal and newborn care interventions for routine care, basic, and comprehensive emergency obstetric and newborn care, and mapped current offerings against these very effectively. There is, however, a lack of clarity as to whether this auditing is the full extent of the authors’ purpose in preparing this manuscript. The abstract and the introductory passages focus on the ‘integration of technical content in training on maternal and newborn health’ (abstract), establishing the expectation that this will be a significant component of the paper. Reading the document, it was unclear if the study was designed as an audit only (with findings confirmed through the qualitative method), or whether there was an initial intent to compare stand-alone and integrated packages and any evidence for their comparative effectiveness.

The abstract suggests that ‘it seems logical and cost-effective to integrate maternal and newborn health trainings’ and this seems to be an important issue for exploration. If this is the focus of the study, this needs to be made clearer throughout the remainder of the document, including highlighting findings on integration of training in the results, and situating these results in broader literature on integrated training and its benefits/ limitations in the discussion. This would help to provide solid, evidence-informed recommendations on what an effective integrated MNH training would look like in Ethiopia and Nepal. The audit of existing training packages provided by this study would also indicate what is in place and what is needed to reach the desired goal of effective integrated training.

If the focus of the paper is on the audit component, and not on the potential of integration of training content and approaches, this should be made clear and the abstract and introductory section re-written to reflect the purpose of the study.

The following comments are divided by manuscript section and include both major and minor issues, and examples of what may need to be addressed. Many of the comments that follow spring from this confusion around the core purpose of the paper.

Title and abstract:

• The title and the abstract do not seem to align. The abstract highlights a focus on integration of training content but this is not mentioned in the title.

Introduction:

• Table 1 is very useful but should be described more fully in the body of the manuscript. It is noted that references are provided at the end of the table, but more is needed in the text about where these essential maternal and newborn care interventions were extracted from, why they were chosen, whether they are the globally accepted standard etc. This is explained on page 6 but needs some explanation here to accompany the table. Explain the key guidelines that informed the construction of this table.

• Page 4: paragraph 1: suggest authors begin paragraph with a broader statement about MNH trainings in general- what is most commonly done- are stand-alone or integrated trainings the norm in LMIC? Are integrated trainings less common? Link this statement back to the opening statement that health workers are often responsible for care across the continuum and so an integrated training- a training that integrates maternal and newborn care would be a logical approach. Maybe also reference the different possible timings of trainings and make clear that this paper is focusing on in-service training, not pre-service etc.

• Page 4 paragraph 2: suggest removing first sentence to page 3 where there is reference to the same individual providing different services- both refer to the work, not the training and it is confusing to put this sentence in the middle of a passage about training.

• Page 4 paragraph 2- more references/evidence are needed throughout the second half of this paragraph. Examples: statements such as ‘…therefore, acquisition of skills may often fall short and quality may be compromised’; ‘There can be a tendency to omit certain topics…’ and others.

• Page 5 paragraph 2: the Kirkpatrick model of training evaluation is mentioned here but not referred to again. Suggest it is unnecessary. It is unclear how the reaction and learning components relate to the results and discussion.

• Pages 5-6 paragraph 2: states that ‘This rapid assessment aims to contribute to the evidence base on differences in training content and implementation approaches for integrated and stand-alone trainings…’ As explained above, this does not clearly come out in the results and discussion. This seems to be an important focus but it is not explored adequately in the sections to follow.

Materials and methods:

• A clear statement of research aims/ objectives/ question(s) would help guide the reader through the remainder of the paper. Again, the abstract highlights a focus on integrated training but this is absent from the title and does not flow through the results and discussion to support recommendations of ways forward for future MNH training packages in these contexts.

Study design

• Suggest changing the order of this section for clarity. Paragraph 1 under study design begins by discussing the identification of relevant training packages- suggest continuing this focus by moving the following passage (abbreviated here) “Available training materials were identified by the senior in-country study coordinator…Essential MNH interventions were based on validated WHO guidelines presented under Table 1” so that it comes directly after “The next step was to identify the national focal person for MNH trainings, and through them, we identified all the relevant national packages that fulfilled our selection criteria”. This allows the reader to better understand the first process of identifying national training packages.

Now that the selection process is more fully described, it is suggested that authors next move on to the review of these selected training packages. This would begin with the passage (abbreviated here) “Training packages were reviewed by two independent…and technical content for routine, basic and comprehensive emergency and obstetric newborn care”. This could then be followed by The extraction tool gave equal weight to all interventions…and not deliberate omissions due to contextual factors” (on page 7).

Adopting this structure means that authors are first describing how training packages were selected, and then describing how they were analysed. Changing the order of text in this way may also require deleting repetition or other editing.

• The inclusion of the data extraction template is excellent. It would be beneficial to provide some information on where this was derived from. If it was from the same documents that informed Table 1, please make this clear.

• Figure 1: provide some example reasons for exclusion.

• Make clear in the body of the manuscript that training packages included for analysis were both stand-alone and integrated.

• The sentence ‘Priority was given to in-service trainings…’ is unclear. Were only in-service trainings included? What was the criteria for this? Were some pre-service trainings included? If so, with what justification?

Data collection

• First paragraph is repeated from previous section.

• Need to be clearer on why the key informants were interviewed. What was the purpose of including this qualitative component? When the research aims/ questions are more clearly defined (as recommended above), the reason for using interviews should be clearly linked and explained. The provided interview guide is very broad so it would be helpful to have a clearer indication of the purpose of these interviews, how this method helped answer the research question, and how it fits with the quantitative analysis/ audit.

Data analysis

• A description of how the 2 components of the research work together to answer the research question is not provided.

Ethics

• Provide an explanation as to why in-country ethics approval was not needed.

Results

• Table 2: It was hard to read with the heavy load of acronyms. Could this be re-formatted to include the names of the training packages within the table?

• The results are well structured as they follow the key components outlined in Table 1.

• The reader can assume the meaning behind the %s given in this section (from p10), but it would be good to explain, at least in the first instance, how percentages were arrived at.

• Page 14- summary of qualitative findings: it would be good to reiterate the rational for conducting these interviews and their focus to guide the reader through the findings.

• From page 15: make clearer that statements included in the text encapsulate what the interviewees stated and are not a reference to other literature or the authors’ own interpretations.

Examples of this:

Pages 14- 15: ‘Implementing such a strategy where specific modules (or trainings) are implemented to address identified gaps in existing quality of care will help to improve knowledge and skills of health workers’. Is this reporting what was said by those interviewed or is this a conclusion drawn by the authors? If it is the former, this should be made clear through reporting verbs such as ‘the respondent explained/ respondents stated…’ etc. If the latter, it should be in the discussion and supported by other evidence.

Page 16: ‘There is a need to update existing…’. If this is what was said by those interviewed, this needs to be clear by adding something like ‘It was reported that…’

Page 16: ‘Rapidly scaling up trainings to meet…’. As above.

Page 17: ‘We need to learn more about effective models for mentorship and supervision…’. As above.

And others- make it clear what was told/ reported to the researchers by the interview participants. If it is interpretation by the authors or relating the responses to other literature, it would be best in the discussion section.

• Some passages seem to be interpretive/ reference other literature/studies and may be best in the discussion:

Examples:

Page 17: ‘In Nepal, program reviews have found that skills deteriorate rapidly…’.

Page 17: ‘Also, skill retention is likely to vary depending on the work…’.

• There is a noted absence of reporting on what participants said in relation to the integration of training packages. This is mentioned in the table e.g., ‘MNH training materials do not cover newborn health…’. As this was a stated focus of the paper, it would be good to have this as a theme if there is sufficient data. This would combine all that was discussed in relation to the overlaps between maternal and newborn health trainings and any comments participants had on this key issue.

Discussion

• Overall, the discussion needs to be improved by reference to existing literature and evidence from other studies or comparable contexts. The discussion does not adequately situate the findings/ results within existing literature and this is especially true for the qualitative findings.

• A deeper discussion section could be provided, followed by recommendations based on the results/ discussion.

• Again, the issue of integration of training contents is not clear in the discussion. If this is a focus of the paper, this needs to be described and the results pertinent to integrated trainings placed within the context of relevant literature. Are there relevant findings from different settings that could be referred to? What is the evidence for integrated training and how do the authors’ findings sit with these? Are there insights from the qualitative data that could provide further clarity here and point to recommendations?

• The discussion also does not elaborate on what the findings/ results mean for integration. Again- if this is the focus of the paper (as indicated by the abstract and the introduction section), this needs to be a focus of the discussion. Where else have integrated training packages been used? What was found in these contexts? What does the data say for Nepal and Ethiopia in this regard and how does this align/ not align with what is known in the literature about the use of stand-alone and integrated training packages? The abstract mentions evidence of benefits that come from integrating training contents but these are not followed through in the results or discussion. More information on this would be helpful in supporting the recommendations.

• If integration is not the focus of the paper, authors should use the discussion to situate the study results (from both the audit and the interviews) more clearly within existing literature on training effectiveness. There is an abundance of resources on training transfer which provide more insight into the nuances of trainees’ capability and willingness to pursue the objectives of their training when they return to work. This literature would be particularly useful to explain and situate the qualitative findings outlined in Table 3 and could be consulted whether integration is the final focus of the manuscript or not.

• There is a lack of reference to existing literature/ referencing in general. Examples, the statements: “Further, babies born to HIV positive mothers tend to be preterm and will need additional feeding and thermal care support…” (p20); and “Similarly, all MNH trainings should cover management of the HIV exposed infant” (p20). These (and others) are without references and do not refer to what is known on these issues from comparable contexts. If these are recommendations arising from this study, they should be stated as such.

• Reference to literature on the recommended alternative approaches to training is also needed (on p21).

6. PLOS authors have the option to publish the peer review history of their article (what does this mean?). If published, this will include your full peer review and any attached files.

Reviewer #1: No

Reviewer #2: No

---

## [Author Response · Author response to Decision Letter 0]

25 Apr 2021

Thank you to the Editor and the peer reviewers for the constructive comments to help improve our paper. This has been very useful for us. Please find the below our line by line responses that highlight the changes we have made to the manuscript based on the comments received from the peer review. Thank you for the opportunity to revise and improve our manuscript and submit to PLOS One. 

Responses to the Editor’s Comments:

Comments E1: (Additional Editor comments) This is an interesting article with potential to make a unique contribution to maternal and newborn health policy, programming and research. Please carefully consider feedback from both reviewers, with particular attention to clarification of research questions/objectives and contextualization of findings within the current global evidence base on the topic of focus.

Response E1: Thank you so much for sharing this reflection. As advised, we have revised the manuscript thoroughly based on the comments received from both peer reviewers. 

Comment E2: Journal Requirements:

Response E2.1: We have reformatted the manuscript in line with the PLoS one style templates. 

2.) Please include additional information regarding the survey or questionnaire used in the study and ensure that you have provided sufficient details that others could replicate the analyses. For instance, if you developed a questionnaire as part of this study and it is not under a copyright more restrictive than CC-BY, please include a copy, in both the original language as well as the English version already provided, as Supporting Information.

Response E2.2: We have attached the questionnaire used for the quantitative data collection in the supplementary section (S1 Questionnaire). 

3.) We note that you have indicated that data from this study are available upon request. PLOS only allows data to be available upon request if there are legal or ethical restrictions on sharing data publicly. For information on unacceptable data access restrictions, please see http://journals.plos.org/plosone/s/data-availability#loc-unacceptable-data-access-restrictions.

 Response E2.3: We have updated the Data Availability statement as “All relevant data are available from within the manuscript as well as a supplemental information file.”

4.) PLOS requires an ORCID iD for the corresponding author in Editorial Manager on papers submitted after December 6th, 2016. Please ensure that you have an ORCID iD and that it is validated in Editorial Manager. To do this, go to ‘Update my Information’ (in the upper left-hand corner of the main menu), and click on the Fetch/Validate link next to the ORCID field. This will take you to the ORCID site and allow you to create a new iD or authenticate a pre-existing iD in Editorial Manager. Please see the following video for instructions on linking an ORCID iD to your Editorial Manager account: https://www.youtube.com/watch?v=_xcclfuvtxQ

 Response E2.4 We have done this. My ORCID id is 0000-0002-8951-3245

Response E2.5 We have corrected this mismatch. We have updated the ‘Funding Information section as follows “This data collected in this study was made possible by the generous support of the American people through the United States Agency for International Development (USAID), under the terms of Cooperative Agreement No. AID-OAA-A-14-00028. The contents are the responsibility of the authors and do not necessarily reflect the views of USAID or the United States Government.”

Responses to the Reviewers' comments:

 Reviewer #1 (R1): 

R1 C1: General comments: The manuscript analyzes the maternal and newborn health training curricula in Ethiopia and Nepal. The authors conducted mixed qualitative and quantitative methods to assess the maternal and newborn health training content and approaches. The authors concluded that the technical content of training curricula had several gaps that should be addressed and the alignment between different MNH training packages should be improved by using innovative methods.

The manuscript needs improvement in grammar, the writing style and the sentence structure. Copy editing is required.

Response R1C1: Thank you so much for this comment. As advised, we have now performed a thorough copyediting of the manuscript and improved language and grammar throughout. 

Abstract:

R1C2: In findings, it is strange to separate active management of the third stage of labor from CmONC as it is one of the signal functions. It would be better to include specific gaps in CmONC.

Response R1C2: Thank you for the comment. The abstract has been now been completey rewritten. 

R1C3: It seems that qualitative findings are missing.

Response R1C3: As advised, we have now added the qualitative findings in the abstract. 

R1C4: The recommendations need to be adjusted based on the aims and findings of the study. The second and third recommendations seem vague. It would be better to come up with a clear and concise conclusion in the abstract.

Response R1C4: Thank you for the comment. We agree and have revised the abstract thoroughly and made the recommendations more specific. “Taken together, our findings suggest four key areas of improvement: first, training materials should be updated based on the content gaps identified and should be aligned with each other. Second, trainings should address actual health worker performance gaps using a variety of innovative approaches such as blended and self-directed learning. Third, post-training supervision and ongoing mentoring needs to be strengthened. Lastly, functional training information systems are required to support planning efforts in both countries.”

Introduction

R1C5: The introduction section needs a fundamental revision. It is a challenge for a reader to follow the sentences and paragraphs. The logical flow between the sentences and paragraphs needs to be considered.

Also, the definition of the authors for this study should be aligned with evidence and literature. Many statements are not supported by references. The rationale for this study is not well described.

Response R1C5: Thank you for your comment. We have revised the introduction section thoroughly. We have ensured all statements are properly referenced and have tried to improve the overall flow of the manuscript and strengthened the rationale for the study. 

R1C6: Page 3: It would better to move the table as an annex and briefly describe the table in the introduction.

Response R1C6: Thank you for this comment. We have now rewritten the introduction section. We feel that the table is better situated in the introduction section since it provides the conceptual framework for our study. We have also added a sentence on page 5, lines 73-76 that describes the contents of the table. However, if you still feel that the table would be better placed in the ‘Supplementary section’, we are happy to move it there. 

R1C7: Page 3, line 6-10: It is hard to understand what the authors are trying to describe here. It does not have a logical flow with the previous sentences.

Response R1C7: We have revised the introduction section and deleted this confusing text. Thank you. 

R1C8: Page 3, table 1: According to proposed new signal functions by Gabrysch et. al BEmONC should be changed to Basic emergency care and CEmONC to Comprehensive emergency care. Otherwise, it creates confusion among readers.

Response R1C8: We have made these changes in revised table 1 as per your suggestion. Thank you. 

R1C9: Page 4, first paragraph: A reference is needed.

Response R1C9: Thank you for your comment. We have revised the introduction section and now added a reference to that statement. Please refer to line 80- page 6. 

R1C10: Page 4, line 6-7: It could be a perception, may not be supported by evidence.

Response R1C10: Thank you. We have now deleted this statement. 

R1C11: Page 4: 13-19: Again, it is a challenge to follow. And the references are missing.

Response R1C11: We have now revised the introduction section, and improved the over all flow. Please see Page 6 lines 83-99. 

R1C12: Page 5, Second paragraph: The logical flow should be considered.

Response R1C12: Thank you. We have deleted the statement about the Kirkpatrick’s model for evaluation of of training effectiveness. 

R1C13: Page 5: The aims of this study should be clear and easy to understand. It is difficult to grasp the aims of the study.

Response R1C13: We have revised the aims statement as follows: “This study aims to audit the training content of existing, government approved MNH training packages and explore the experiences of the stakeholders regarding the implementation of these training packages in Ethiopia and Nepal. Please see page 8 Lines 133-136. 

R1C14: Page 5: Line 18-21: This sentence needs to be moved to the acknowledgments.

Response R1C14: We have removed the sentence from the manuscript altogether. 

R1C15: Page 5: The last paragraph should be integrated with the aims of the study followed by possible implications.

Response R1C15: Thank you for your comment. We have now revised this section as follows. “In addition, we validated our findings with national experts and stakeholders in both countries and jointly developed recommendations for strengthening in-service trainings for maternal and newborn health in Ethiopia and Nepal.” Please see lines 134-136 on page 9. 

Methods:

R1C16: Again, the logical flow of the ideas should seriously be considered. It’s a challenge for a reader to follow the jumping ideas in these sentences. It would be better to follow a smooth transition from one sentence to another and clarify the linkages.

Response R1C16: Thank you for your comment. We have now restructured the entire methods section and signposted with clear sub-headings. 

R1C17: Page 7: The last two paragraphs are relevant to the introduction section.

Response R1C17: We have moved these to the introduction section. (page 8, Lines 113-126.)

R1C18: The authors need to describe Figure 1. the essential data in the text and clearly explain the inclusion and exclusion criteria.

Response R1C18: Thank you. We have now clearly explained the inclusion and exculsion criteria used for the training curricula audit. Please see page 9-10, lines 152-161. 

R1C19: “Priority was given to in-service training and interventions at health facility setting only.” It is a standalone sentence without any justification or profound description.

Response R1C19: We have now removed this sentence since our review only looks at in-service trainings. 

R1C20: The tools development process and justification should further be explained.

Response R1C20: Thank you. We have made a separate sub-section for ‘Study tools’ and have described the development of the study tool. Please refer to page 11, lines 186-196. 

R1C21: It would be better to describe the type of software used for qualitative data analysis and the quality assurance data analysis- intercoder reliability assessment.

Response R1C21: Thank you for this comment. This work was undertaken was as a rapid assessment to strengthen MOH’s ongoing programs and not as a focused qualitative study. We used Microsoft excel to analyze the qualitative data. A thematic approach was used where two reserachers jointly reviewed the interview transcripts line and by line and agreed on the sets of codes. Both researchers then jointly coded all the open-ended comments. In cases where disagreements arose between researchers, further discussion took place until consensus was achieved. Throughout the analysis process, researchers reflected on how their background, training and worldview might influence their interpretation of results and efforts were taken to minimise them. This data was also validated with a group of national experts in both countries. We triangulated the quantitative data with qualitative findings. Comments that summarise common findings across observations are reported.In addition, we have acknowledged the possibility of interviewer bias in the limitations section as well. 

Result:

R1C22: The authors need to consider reorganizing the results section and retain the consistency of presenting the findings. It would be better to include numbers beside the proportion for each clinical practice as number (%).

Response R1C22: Thank you for this comment. As advised, we have revised the results section and tried to maintain a consistent approach to present the results. For presentation purposes and given the word count limitations, we have presented actual proportions only. We feel that including absolute numbers together with the proportions may make it hard for readers to follow. In the data analysis section, we have clarified that the proportion for each clinical practice was calculated as the total number of YES responses divided by the total number of interventions in that practice. If the reviewer and editor still want the us to include the absolute numbers, we would be happy to do so. Please advise. 

R1C23: Page 11: Why did authors present the findings of different packages such as PMTCT, NICU, and IMNCI under the subheading of Newborn care interventions in BEmONC and CEmOC. These are confusing! If the authors are following Sabine Gabrysch et.al proposed framework, then the terms need to be clarified as basic or comprehensive emergency care to avoid confusion.

Response R1C23:Thanks for your comment. We have now revised the subheadings as basic emergency care and comprehensive emergency care through out the manuscript in line with the Gabrysch et.al framework. 

R1C24: Presenting the data is not consistent throughout the results section e.g. page 12.

Response R1C24: Thank you for this comment. As advised, we have revised the results section and tried to maintain a consistent approach to present the results throughout. We have now added the percentages (%) in several places where they were missing. 

R1C25: Page 12: Again, “In Ethiopia, for CEmONC, fluid management in a newborn, safe oxygen therapy and b-CPAP therapy were covered comprehensively in the NICU training materials,”. It is confusing to see the CEmONC is mixed up with different training packages. It seems that BEmONC and EmONC packages and signal functions are miss-interpreted in the results section.

Response R1C25: Thank you for your comment. We have now used ‘Comprehensive emergency care’ and ‘Basic emergency care’ to refer to the framework and only used CEmONC and BEmONC to refer to the training packages used in Ethiopia which have the same name. 

R1C26: Page 13: “the national BEmONC manual was found to be comprehensive”. It seems subjective without presenting the data.

Response R1C26: This sentence has been updated as follows: In Ethiopia, the national BEmONC manual was complete with all clinical interventions duly reflected (100%). Please refer to Page 19 Line 312-313. We have now added the percentages (%) in this sentence and everywhere relevant throughout the results. 

R1C27: The authors don’t need to present all the details of the findings in the qualitative section. They can focus more on the main and key findings. They also need to include some quotes from the participants of the study to support the findings and make the results more interesting. It would be interesting to see subheadings based on the themes.

Response R1C27: Thank you for your comment. We have now arranged the qualitative findings section and provided subheadings of each theme and added some quotes. (Page 24, Line 420-422) 

Discussion:

R1C28: The findings which are presented in the results section need to be interpreted in the discussion section but not the other new findings. For example: ”Specifically, newborn infection prevention guidelines were found to be deficient for topics such as hand washing before and after handling babies, the importance of rooming-in with the mother,

benefits of co-bedding, encouragement for early breastfeeding...". It would be strange to see these results for the first time.

Response R1C28: Thank you. We have now removed these findings from the discussion section. 

R1C29: It would be better to discuss the qualitative results, not only the recommendations from the participants.

Response R1C29: Thank you for your comment. We have now strengthened the discussion of the qualitative section. 

Conclusion:

R1C30: The first paragraph seems unnecessary. The first sentence of the second paragraph is general. It would be better to make it more specific.

Response R1C30:Thank you for your comment. We have now removed the introductory paragraph of the conclusions section which was generic. Please refer to lines 535 onwards on page 29. 

Reviewer #2 (R2): 

R2C1: This is an important study with the potential to help us understand gaps in the delivery of maternal and newborn health training content and approaches in Ethiopia, Nepal and more broadly. The paper is well written with only some minor grammatical edits required. The study sought to audit existing training packages in the two contexts of interest, and the authors have effectively defined essential maternal and newborn care interventions for routine care, basic, and comprehensive emergency obstetric and newborn care, and mapped current offerings against these very effectively. There is, however, a lack of clarity as to whether this auditing is the full extent of the authors’ purpose in preparing this manuscript. The abstract and the introductory passages focus on the ‘integration of technical content in training on maternal and newborn health’ (abstract), establishing the expectation that this will be a significant component of the paper. Reading the document, it was unclear if the study was designed as an audit only (with findings confirmed through the qualitative method), or whether there was an initial intent to compare stand-alone and integrated packages and any evidence for their comparative effectiveness.

Response R2C1: Thank you for your comment. We have now delected the emphasis on integration in the abstract and introductory passages and clearly spelled out our aims of the study. We have audited the contents of different training packages and explored the expriences of stakeholders in implementing these training packages. Please refer toPage 8-9, Lines 132-134. 

R2C2: The abstract suggests that ‘it seems logical and cost-effective to integrate maternal and newborn health trainings’ and this seems to be an important issue for exploration. If this is the focus of the study, this needs to be made clearer throughout the remainder of the document, including highlighting findings on integration of training in the results, and situating these results in broader literature on integrated training and its benefits/ limitations in the discussion. This would help to provide solid, evidence-informed recommendations on what an effective integrated MNH training would look like in Ethiopia and Nepal. The audit of existing training packages provided by this study would also indicate what is in place and what is needed to reach the desired goal of effective integrated training.

If the focus of the paper is on the audit component, and not on the potential of integration of training content and approaches, this should be made clear and the abstract and introductory section re-written to reflect the purpose of the study.

The following comments are divided by manuscript section and include both major and minor issues, and examples of what may need to be addressed. Many of the comments that follow spring from this confusion around the core purpose of the paper.

Response R2C2: Thank you. This is very useful comment. We have now revised the abstract and the introduction sections of the manuscript. As mentioned earlier, the aims of the paper have now been clearly articulated. This study aims to audit the training content of existing, government-approved MNH training packages and explore the experiences of stakeholders regarding the implementation of these training packages in Ethiopia and Nepal. Our intention is not to compare standalone or integrated training packages in either of the countries. 

Title and abstract:

R2C3: The title and the abstract do not seem to align. The abstract highlights a focus on integration of training content but this is not mentioned in the title.

Response R2C3: 

We have revised the abstract and rewritten significant portions of the manuscript. There is now better flow and logic throughout the manuscript. 

Introduction:

R2C4: Table 1 is very useful but should be described more fully in the body of the manuscript. It is noted that references are provided at the end of the table, but more is needed in the text about where these essential maternal and newborn care interventions were extracted from, why they were chosen, whether they are the globally accepted standard etc. This is explained on page 6 but needs some explanation here to accompany the table. Explain the key guidelines that informed the construction of this table.

Response R2C4: Thank you for this comment. We have now added some text about essential maternal and newborn care interventions listed in Table 1. Please see lines 72-78 in page 5 and 6. 

R2C5: • Page 4: paragraph 1: suggest authors begin paragraph with a broader statement about MNH trainings in general- what is most commonly done- are stand-alone or integrated trainings the norm in LMIC? Are integrated trainings less common? Link this statement back to the opening statement that health workers are often responsible for care across the continuum and so an integrated training- a training that integrates maternal and newborn care would be a logical approach. Maybe also reference the different possible timings of trainings and make clear that this paper is focusing on in-service training, not pre-service etc.

Response R2C5:

Thank you for this comment. We have extensively rewritten the introduction section now and tried to clarify that our focus is on in-services trainings and that health workers provide care across the continuum of care. Please refer to page 6 line 82 onwards. 

R2C6: • Page 4 paragraph 2: suggest removing first sentence to page 3 where there is reference to the same individual providing different services- both refer to the work, not the training and it is confusing to put this sentence in the middle of a passage about training.

Response R2C6: Thank you for this comment. We have now rewritten the introduction section and strengthened it. 

R2C7: • Page 4 paragraph 2- more references/evidence are needed throughout the second half of this paragraph. Examples: statements such as ‘…therefore, acquisition of skills may often fall short and quality may be compromised’; ‘There can be a tendency to omit certain topics…’ and others.

Response R2C7: Thank you for this comment. We have now strengthened the introduction section and added some additional references that have tried to examine training related inputs for improving MNH. We have also deleted some of the sentences that did not have strong research evidence. 

R2C8: • Page 5 paragraph 2: the Kirkpatrick model of training evaluation is mentioned here but not referred to again. Suggest it is unnecessary. It is unclear how the reaction and learning components relate to the results and discussion.

Response R2C8: Thank you. We have removed this in the revised manuscript. 

R2C9: • Pages 5-6 paragraph 2: states that ‘This rapid assessment aims to contribute to the evidence base on differences in training content and implementation approaches for integrated and stand-alone trainings…’ As explained above, this does not clearly come out in the results and discussion. This seems to be an important focus but it is not explored adequately in the sections to follow.

Response R2C9: 

We have now extensively revised the manuscript including refining the aims of the study. We have also removed any mention of making comparision between the benefits and pitfalls of integrated versus standalone approaches through out the manuscript. Hope this is sufficient. Please advise if any further changes are needed. 

Materials and methods:

R2C10:• A clear statement of research aims/ objectives/ question(s) would help guide the reader through the remainder of the paper. Again, the abstract highlights a focus on integrated training but this is absent from the title and does not flow through the results and discussion to support recommendations of ways forward for future MNH training packages in these contexts.

Response R2C10: We have now clarified our aim as “This study aims to audit the training content of existing, government-approved MNH training packages and explore the experiences of the stakeholders regarding the implementation of these training packages in Ethiopia and Nepal.”. The manuscript has also been extensively rewritten and reads much better. 

Study design

R2C11: • Suggest changing the order of this section for clarity. Paragraph 1 under study design begins by discussing the identification of relevant training packages- suggest continuing this focus by moving the following passage (abbreviated here) “Available training materials were identified by the senior in-country study coordinator…Essential MNH interventions were based on validated WHO guidelines presented under Table 1” so that it comes directly after “The next step was to identify the national focal person for MNH trainings, and through them, we identified all the relevant national packages that fulfilled our selection criteria”. This allows the reader to better understand the first process of identifying national training packages.

Now that the selection process is more fully described, it is suggested that authors next move on to the review of these selected training packages. This would begin with the passage (abbreviated here) “Training packages were reviewed by two independent…and technical content for routine, basic and comprehensive emergency and obstetric newborn care”. This could then be followed by The extraction tool gave equal weight to all interventions…and not deliberate omissions due to contextual factors” (on page 7).

Adopting this structure means that authors are first describing how training packages were selected, and then describing how they were analysed. Changing the order of text in this way may also require deleting repetition or other editing.

Response R2C11: Thank you for the suggestion. We have made all the suggested changes throughout the manuscript and also added subheadings in the methods section to improve the flow. 

R2C12: • The inclusion of the data extraction template is excellent. It would be beneficial to provide some information on where this was derived from. If it was from the same documents that informed Table 1, please make this clear.

Response R2C12: Thank you. We have clarified this in the revised manuscript that the tool was based on the validated WHO guidelines and the conceptual framework as presented in table 1. (Page 11, Line 188 onwards) 

R2C13: • Figure 1: provide some example reasons for exclusion.

Response R2C13: We have provided the reasons for exclusion and inclusion in the study methods section (Line 156-160) and the overall study flowchart in Figure 1 also highlights this. 

R2C14: • Make clear in the body of the manuscript that training packages included for analysis were both stand-alone and integrated.

Response R2C14: We have added a sentence in the study design section to clarify that we have included both stand-alone and integrated trainings in our analysis. 

R2C15: • The sentence ‘Priority was given to in-service trainings…’ is unclear. Were only in-service trainings included? What was the criteria for this? Were some pre-service trainings included? If so, with what justification?

Response R2C15: Thank you for the suggestion. We have removed this confusing statement. To clarify, all the trainings we have included in our analysis are in-service trainings. 

Data collection

R2C16:• First paragraph is repeated from previous section.

Response R2C16: We have removed this paragraph now. 

R2C17: • Need to be clearer on why the key informants were interviewed. What was the purpose of including this qualitative component? When the research aims/ questions are more clearly defined (as recommended above), the reason for using interviews should be clearly linked and explained. The provided interview guide is very broad so it would be helpful to have a clearer indication of the purpose of these interviews, how this method helped answer the research question, and how it fits with the quantitative analysis/ audit.

Response R2C17: Thank you for the comment. Based on both reviewers comments, we have rewritten the aims of the study as follows. “This study aims to audit the training content of existing, government-approved MNH training packages and explore the experiences of the stakeholders regarding the implementation of these training packages in Ethiopia and Nepal. “ The purpose of qualitative interviews was to generate a greater contextual understanding of how future traionings could be strengthened. 

Data analysis

R2C18: • A description of how the 2 components of the research work together to answer the research question is not provided.

Response R2C18: We have added a paragraph at the end of the Data analysis section explaining the value of using the mixed-methods approach in our paper. Please refer to Page 12, Lines 218-223. 

Ethics

R2C19: • Provide an explanation as to why in-country ethics approval was not needed.

Response R2C19: This assessment was approved by the Save the Children’s Ethical Review board and the in-country assessment was led by the Ministry of Health in each country. The MOH approved the workplan in the country and were involved in conducting the study in both countries. To clarify, this activity was not planned as a separate/ dedicated research activity but was conducted as a part of ongoing training implementation, led by the MOH and supported by partners to strengthen existing national in-service MNH trainings in both countries. 

Results

R2C20: • Table 2: It was hard to read with the heavy load of acronyms. Could this be re-formatted to include the names of the training packages within the table?

Response R2C20: We have now reformatted the table in landscape mode and provided full forms for all of the acronyms. (Page 14-15) 

R2C21: • The results are well structured as they follow the key components outlined in Table 1.

Response R2C21: Thank you so much. 

R2C22: • The reader can assume the meaning behind the %s given in this section (from p10), but it would be good to explain, at least in the first instance, how percentages were arrived at.

Response R2C22: Thank you. We have now clarified this in the Data analysis section. Please see Page 12, lines 204-209. 

R2C23: • Page 14- summary of qualitative findings: it would be good to reiterate the rational for conducting these interviews and their focus to guide the reader through the findings.

Response R2C23: Thank you. We have added a sentence right at the beginning before we summarise the qualitative findings as suggested. Please see page 20- lines 342-344. 

From page 15: make clearer that statements included in the text encapsulate what the interviewees stated and are not a reference to other literature or the authors’ own interpretations.

Examples of this:

R2C24: Pages 14- 15: ‘Implementing such a strategy where specific modules (or trainings) are implemented to address identified gaps in existing quality of care will help to improve knowledge and skills of health workers’. Is this reporting what was said by those interviewed or is this a conclusion drawn by the authors? If it is the former, this should be made clear through reporting verbs such as ‘the respondent explained/ respondents stated…’ etc. If the latter, it should be in the discussion and supported by other evidence.

Response R2C24: This was expressed by the participants, so we have clarified that. Plase see Page 22 Lines 358-360. 

R2C25: Page 16: ‘There is a need to update existing…’. If this is what was said by those interviewed, this needs to be clear by adding something like ‘It was reported that…’

Response R2C25: We have made it clear that it was reported as expressed by the participants. (Page 22, Lines 392-394)

R2C26: Page 16: ‘Rapidly scaling up trainings to meet…’. As above.

Response R2C26: We have made it clear that it was reported as expressed by the participants. (Page 24, Lines 409-412) 

R2C27: Page 17: ‘We need to learn more about effective models for mentorship and supervision…’. As above.

Response R2C27: We have made it clear that it was reported as expressed by the participants. (Page 24, Lines 423-425)

R2C28: And others- make it clear what was told/ reported to the researchers by the interview participants. If it is interpretation by the authors or relating the responses to other literature, it would be best in the discussion section. 

Response.R2C28: Thank you. We have now clarified throughout the discussion that participants had expressed these issues and that it is not our interpretation. 

Some passages seem to be interpretive/ reference other literature/studies and may be best in the discussion: Examples:

R2C29: Page 17: ‘In Nepal, program reviews have found that skills deteriorate rapidly…’.

Response R2C29: This was interpretative and also supported by a reference. We have moved this to the appropriate section in the discussion. (Page 28, Line 507)

R2C30: Page 17: ‘Also, skill retention is likely to vary depending on the work…’.

Response R2C30: Thank you for this comment. This was interpretative and was referenced using a report. We have moved this to the appropriate section in the discussion. (Page 28, Line 509)

R2C31: • There is a noted absence of reporting on what participants said in relation to the integration of training packages. This is mentioned in the table e.g., ‘MNH training materials do not cover newborn health…’. As this was a stated focus of the paper, it would be good to have this as a theme if there is sufficient data. This would combine all that was discussed in relation to the overlaps between maternal and newborn health trainings and any comments participants had on this key issue.

 Response R2C31: Thanks for this comment. We have a subheading on ‘Limited technical content for newborn health’ while discussing the qualitative findings. Please refer to Page 23, lines 391 to 402. 

Discussion

R2C32: • Overall, the discussion needs to be improved by reference to existing literature and evidence from other studies or comparable contexts. The discussion does not adequately situate the findings/ results within existing literature and this is especially true for the qualitative findings. A deeper discussion section could be provided, followed by recommendations based on the results/ discussion.

Response R2C32: Thank you for this comment. We have added additional references to better situate our findings with relate existing body of work. Please refer to references 38, 41,42,43,45,46,47,50 and 51 in page 33 and 34. 

R2C33: • Again, the issue of integration of training contents is not clear in the discussion. If this is a focus of the paper, this needs to be described and the results pertinent to integrated trainings placed within the context of relevant literature. Are there relevant findings from different settings that could be referred to? What is the evidence for integrated training and how do the authors’ findings sit with these? Are there insights from the qualitative data that could provide further clarity here and point to recommendations?

Response R2C33: We have now rewritten the abstract, most of the introduction and discussion sections to clarify that this manuscript only reports on findings from audit of different training pacakges. 

R2C34: • The discussion also does not elaborate on what the findings/ results mean for integration. Again- if this is the focus of the paper (as indicated by the abstract and the introduction section), this needs to be a focus of the discussion. Where else have integrated training packages been used? What was found in these contexts? What does the data say for Nepal and Ethiopia in this regard and how does this align/ not align with what is known in the literature about the use of stand-alone and integrated training packages? The abstract mentions evidence of benefits that come from integrating training contents but these are not followed through in the results or discussion. More information on this would be helpful in supporting the recommendations.

Response R2C34: Thank you for this comment. We have now siginificantly revised the manuscript and strengthened our arguments by providing additional references throughout the manuscript. We have also made it clear that our intention is not to make any comparisons to standalone versus integrated trainings. 

R2C35: • If integration is not the focus of the paper, authors should use the discussion to situate the study results (from both the audit and the interviews) more clearly within existing literature on training effectiveness. There is an abundance of resources on training transfer which provide more insight into the nuances of trainees’ capability and willingness to pursue the objectives of their training when they return to work. This literature would be particularly useful to explain and situate the qualitative findings outlined in Table 3 and could be consulted whether integration is the final focus of the manuscript or not.: 

Response R2C35: Thank you for your excellent suggestions! We have now added additional refences to enrich the discussion section and rewritten the discussion particularly for the qualitative results presented under Table 3.

R2C36: • There is a lack of reference to existing literature/ referencing in general. Examples, the statements: “Further, babies born to HIV positive mothers tend to be preterm and will need additional feeding and thermal care support…” (p20); and “Similarly, all MNH trainings should cover management of the HIV exposed infant” (p20). These (and others) are without references and do not refer to what is known on these issues from comparable contexts. If these are recommendations arising from this study, they should be stated as such.

Response R2C36: Thanks for this comment. We have now added two references to support our statement that HIV positive mother are likelier to have increased odds of preterm birth. To clarify, our argument is that, if the odds of preterm births are higher amongst HIV positive women, then health workers caring for these women and their newborns, should also be trained on providing additional feeding support and thermal care . Hence, we feel that feel that MNH manuals should cover these issues as well. Globally, the complications of preterm birth are now the most important cause of newborn deaths and prevalence of preterm birth in Ethiopia is 10.5% and Nepal is 9.3%, so this is an important finding for these contexts. (Page 26, Lines 464-471) 

R2C37: • Reference to literature on the recommended alternative approaches to training is also needed (on p21).

Response R2C37: Thank you. We have now added some references and strengthened this recommendation as well. Please see Page 27, Lines 494-498.

---

## [Decision Letter · Decision Letter 1]

3 Aug 2021

PONE-D-20-38462R1

Analysis of maternal and newborn health training content and approaches to inform future training programs for maternal and newborn care in the low- and middle-income countries: lessons from Ethiopia and Nepal

PLOS ONE

Dear Dr. Sharma,

Thank you for submitting your manuscript to PLOS ONE. After careful consideration, we feel that it has merit but does not fully meet PLOS ONE’s publication criteria as it currently stands. Therefore, we invite you to submit a revised version of the manuscript that addresses the points raised during the review process.

We look forward to receiving your revised manuscript.

Kind regards,

Hannah Tappis, DrPH, MPH

Academic Editor

PLOS ONE

Journal Requirements:

Additional Editor Comments (if provided):

Please carefully consider feedback from both reviewers, with particular attention to outstanding questions regarding methodology and risk of biases or subjectivity in presentation of results. A comprehesnive copyedit is also recommeded.

Reviewers' comments:

Reviewer's Responses to Questions

**Comments to the Author**

1. If the authors have adequately addressed your comments raised in a previous round of review and you feel that this manuscript is now acceptable for publication, you may indicate that here to bypass the “Comments to the Author” section, enter your conflict of interest statement in the “Confidential to Editor” section, and submit your "Accept" recommendation.

Reviewer #1: (No Response)

Reviewer #3: (No Response)

2. Is the manuscript technically sound, and do the data support the conclusions?

Reviewer #1: Yes

Reviewer #3: Yes

3. Has the statistical analysis been performed appropriately and rigorously? 

Reviewer #1: Yes

Reviewer #3: Yes

4. Have the authors made all data underlying the findings in their manuscript fully available?

Reviewer #1: Yes

Reviewer #3: No

5. Is the manuscript presented in an intelligible fashion and written in standard English?

Reviewer #1: No

Reviewer #3: Yes

6. Review Comments to the Author

Reviewer #1: The authors have brought considerable changes to the manuscript and incorporated previous comments and suggestions. Below suggestions will help to improve the manuscript further.

There are some grammatical and typographical errors in the manuscript, and the authors need to correct them.

Abstract:

The aim of the study seems complex. Therefore, the authors need to make the aim of the study simple, concise, and focused. Meanwhile, the integration of training packages and services stated in the first paragraph does not match the study's aim.

Keywords: It would be better to match the selected keywords with MeSH.

Introduction:

The introduction section is unnecessarily lengthy. It would be better to keep this section brief, clear, and aligned with the objective of the study. Several paragraphs are slipping away from the study's central concern, e.g., page 5, second paragraph. In addition, the aim of the study stated in the introduction should be aligned with the abstract.

Methods:

The authors need to include relevant references to the selected design/methods of the study.

Introduction.

The introduction section is unnecessarily lengthy. It would be better to keep this section brief, clear, and aligned with the study's objective. Several paragraphs are slipping away from the study's central concern, e.g., page 5, second paragraph. In addition, the aim of the study stated in the introduction should be aligned with the abstract.

Methods:

The authors need to include relevant references to the selected design/methods of the study.

Results:

Figures 2 and 3 captions should be adjusted to the subheadings (page 16-18). Adding some quotes from the study participants will enrich the qualitative results.

Discussion:

It will be better to compare the findings with other similar studies in LMIC and explain the differences.

Reviewer #3: I commend the authors for addressing a topic that has large programmatic relevance. Maternal and newborn health is an integral part of sustainable development targets. Huge amount of money government and aid money is spent on MNH training.

Generally speaking, the manuscript is written well and followed a sound technical approach. However, there are a few areas that need explanation and/or improvement.

1. The authors stated that their evaluation targeted levels 1 (reaction) and 2 (learning) of Kirkpatrick Framework but I have not seen the learning component clearly assessed in the paper. The authors should review the introduction section where they describe the Kirkpatrick evaluation framework. Levels 2 and 3 are exchanged.

2. In the methods section, the authors stated that they excluded training packages that are not approved by the MOH in the focus countries. What proportion of MNH trainings followed approved training materials? How might this have affected the selection and results?

3. The data extraction tool has binary responses (Yes or No). I wonder if the data extractors had difficulty in giving a Yes/No answer and what influence that might have had on the findings.

4. The manuscript identified missing technical contents in the reviewed materials. On the other hand, the authors rightly acknowledge the concerns about integration and said that training should respond to specific performance gaps instead of universal coverage of topics. The latter suggests we should not expect every training material to cover every related topic. It is not clear how the authors determined essential from nice to know content. Lack of clarity on this can potentially affect validity of conclusions.

There are needs for editorial improvements. Full stop at the end of sentences is missing in many places.

7. PLOS authors have the option to publish the peer review history of their article (what does this mean?). If published, this will include your full peer review and any attached files.

Reviewer #1: No

Reviewer #3: **Yes: **Tegbar Yigzaw Sendekie

---

## [Author Response · Author response to Decision Letter 1]

21 Sep 2021

Response to reviewers

Reviewer #1: The authors have brought considerable changes to the manuscript and incorporated previous comments and suggestions. The below suggestions will help to improve the manuscript further.

Response: Thank you to the reviewers for your thoughtful and constructive comments which have helped to strengthen our manuscript further. Please find our responses below. 

Reviewer 1, Comment 1: There are some grammatical and typographical errors in the manuscript, and the authors need to correct them.

Reviewer 1, Response 1: We have reviewed and revised the manuscript thoroughly for grammatical and typological errors. 

Abstract:

Reviewer 1, Comment 2: The aim of the study seems complex. Therefore, the authors need to make the aim of the study simple, concise, and focused. Meanwhile, the integration of training packages and services stated in the first paragraph does not match the study's aim.

Reviewer 1, Response 2:

We have now revised the abstract to make it simpler as suggested. 

Reviewer 1, Comment 3: Keywords: It would be better to match the selected keywords with MeSH.

Reviewer 1, Response 3:We have now matched keywords with MeSH headings as suggested. 

Developing Countries, Health Services, Maternal Health, Newborn Health, Training, Inservice, On-the-Job, Ethiopia, Nepal. 

Introduction:

Reviewer 1, Comment 4: The introduction section is unnecessarily lengthy. It would be better to keep this section brief, clear, and aligned with the objective of the study. Several paragraphs are slipping away from the study's central concern, e.g., page 5, second paragraph. In addition, the aim of the study stated in the introduction should be aligned with the abstract.

Reviewer 1, Response 4: We have now revised the introduction section, tightened the text and deleted repetitions. We have also aligned the aims of the study in the introduction section with the abstract as advised. 

Methods:

Reviewer 1, Comment 5: The authors need to include relevant references to the selected design/methods of the study.

Reviewer 1, Response 5: Thanks. We have now added two references for the methods/ design section. 

Results:

Reviewer 1, Comment 6: Figures 2 and 3 captions should be adjusted to the subheadings (page 16-18). 

Reviewer 1, Response 6: We have adjusted the captions accordingly as suggested. 

Reviewer 1, Comment 7: Adding some quotes from the study participants will enrich the qualitative results.

Reviewer 1, Response 7: Thank you for your comment. We did not put quotations initially given the word count but have now added two quotations on page 24 lines 439 onwards. 

Discussion:

Reviewer 1, Comment 8: It will be better to compare the findings with other similar studies in LMIC and explain the differences.

Reviewer 1, Response 8: To the best of our knowledge, there is limited research evidence from peer-reviewed and grey literature systematically assessing training content and implementation experiences of in-services training programs in Nepal and Ethiopia. We have added the following to line 551 in the discussion section. This study adds to the limited but growing evidence-base on the content of various in-service training materials and their implementation experience in both countries. 

Reviewer #3: I commend the authors for addressing a topic that has large programmatic relevance. Maternal and newborn health is an integral part of sustainable development targets. A huge amount of government and aid money is spent on MNH training.

Generally speaking, the manuscript is written well and followed a sound technical approach. However, there are a few areas that need explanation and/or improvement.

Reviewer 3, Comment 1: The authors stated that their evaluation targeted levels 1 (reaction) and 2 (learning) of the Kirkpatrick Framework but I have not seen the learning component clearly assessed in the paper. The authors should review the introduction section where they describe the Kirkpatrick evaluation framework. Levels 2 and 3 are exchanged.

Reviewer 3, Response 1: Kindly note that the reference to the Kirkpatrick framework was removed during the first revision of the manuscript. Hence, we can confirm that this version of the manuscript has no mention of and does not refer to the Kirkpatrick framework. 

Reviewer 3, Comment 2: In the methods section, the authors stated that they excluded training packages that are not approved by the MOH in the focus countries. What proportion of MNH trainings followed approved training materials? How might this have affected the selection and results?

Reviewer 3, Response 2: Our selection criteria for inclusion was limited to collecting and analyzing existing national-level training materials which were endorsed and approved by the MoH in both countries. We did not capture information on the whole spectrum of training materials that may exist in countries and limited ourselves to those that were officially endorsed or approved by the MOH for implementation. Therefore, we are not in a position to provide any insights on other “non-approved” training materials. In addition, we validated our findings with program focal persons and key stakeholders in both countries during national consultative workshops (attended by >20 participants) and are confident that we have included all the relevant training materials. 

Reviewer 3, Comment 3: The data extraction tool has binary responses (Yes or No). I wonder if the data extractors had difficulty in giving a Yes/No answer and what influence that might have had on the findings.

Reviewer 3, Response 3:

The lead researchers in both countries were experienced senior technical experts and did not have any difficulty in assigning yes/ no responses. Wherever technical content was found to be lacking compared to our data extraction tool, researchers gave that a no response. In addition, the key informant interviews and the validation workshops were used to clarify any technical nuances or specific clinical details. 

Reviewer 3, Comment 4: The manuscript identified missing technical contents in the reviewed materials. On the other hand, the authors rightly acknowledge the concerns about integration and said that training should respond to specific performance gaps instead of universal coverage of topics. The latter suggests we should not expect every training material to cover every related topic. It is not clear how the authors determined essential from nice to know content. Lack of clarity on this can potentially affect the validity of conclusions.

Reviewer 3, Response 4: Thanks for this very important comment. The recommendations that emerged from this study respond to the gaps identified from the desk review, key informant interviews and consultations with national experts. On the issue of the technical scope of the training materials, our recommendations are to close down identified gaps based on the minimum technical standards or signal functions that have been identified for routine care, basic and comprehensive obstetric and neonatal care recommended by the World Health organization and other technical agencies. These signal functions are the most effective medical interventions that are essential for managing maternal and newborn health complications and will require trained health workers working in appropriately resourced facilities to provide high-quality care to the population. 

Reviewer 3, Comment 5: There are needs for editorial improvements. Full stop at the end of sentences is missing in many places.

Reviewer 3, Response 5: Thank you. We have reviewed and revised the manuscript thoroughly for grammatical and typological errors including punctuations.

---

## [Editor Report · Decision Letter 2]

4 Oct 2021

Analysis of maternal and newborn training curricula and approaches to inform future trainings for routine care, basic and comprehensive emergency obstetric and newborn care in the low- and middle-income countries: lessons from Ethiopia and Nepal

PONE-D-20-38462R2

Dear Dr. Sharma,

We’re pleased to inform you that your manuscript has been judged scientifically suitable for publication and will be formally accepted for publication once it meets all outstanding technical requirements.

Kind regards,

Hannah Tappis, DrPH, MPH

Academic Editor

PLOS ONE

Additional Editor Comments (optional): Please ensure a careful copyedit at the proof stage. There are a few minor language/style issues remaining (e.g. 'mix methods' used instead of 'mixed methods').
---

## [Editor Report · Acceptance letter]

11 Oct 2021

PONE-D-20-38462R2 

Analysis of maternal and newborn training curricula and approaches to inform future trainings for routine care, basic and comprehensive emergency obstetric and newborn care in the low- and middle-income countries: lessons from Ethiopia and Nepal 

Dear Dr. Sharma:

I'm pleased to inform you that your manuscript has been deemed suitable for publication in PLOS ONE. Congratulations! Your manuscript is now with our production department. 

Kind regards, 

on behalf of

Dr. Hannah Tappis 

Academic Editor

PLOS ONE